# RELATIONAL LEARNING WITH VARIATIONAL BAYES

## ABSTRACT

In psychology, relational learning refers to the ability to recognize and respond to relationship among objects irrespective of the nature of those objects. Relational learning has long been recognized as a hallmark of human cognition and a key question in artificial intelligence research. In this work, we propose an unsupervised learning method for addressing the relational learning problem where we learn the underlying relationship between a pair of data irrespective of the nature of those data. The central idea of the proposed method is to encapsulate the relational learning problem with a probabilistic graphical model in which we perform inference to learn about data relationships and other relational processing tasks.

## 1 INTRODUCTION

American Psychological Association defines *relational learning* as (VandenBos & APA, 2007):

**Definition 1.1** (**Relational learning**). *Learning to differentiate among stimuli on the basis of relational properties rather than absolute properties.*

In other words, relational learning refers to the ability to recognize and respond to relationship (called *relational property*) among objects irrespective of the nature of those objects (called *absolute property*). Relational learning has long been recognized as a hallmark of human cognition, and there has been substantial research showing that adequate cognitive capacity is necessary for relational processing (Biederman, 1987; Medin et al., 1993; Holyoak, 2012; Doumas & Hummel, 2013; Gentner, 2016). As a machine learning application, relational learning can provide new insight into data analysis by dissecting information in the data into relational property and absolute property. However, in order to discover relationship patterns among raw and unknown data, relational learning is only truly useful if it can be achieved without supervised data. A key challenge in learning relational property with machine learning-based methods is that relational property is an abstract construct; unlike absolute property, which is based on observable data and can be quantitatively measured, relational property is an abstract quantity that is difficult to objectively quantify, especially when the learning is unsupervised.

In this work, we propose an unsupervised learning method—variational relation learning (VRL)—for addressing the relational learning problem. The proposed method is completely unsupervised, which means that the learning does not require a labeled training dataset nor training examples that have the same (or different) relational property. At its core, VRL encapsulates the relational learning problem with a probabilistic graphical model (PGM) in which we perform inference to learn about relational property and other relational processing tasks. Furthermore, our main learning algorithm is derived from the PGM using first principles, which gives us the flexibility to use any compatible computational inference method and still retains the desired properties of the proposed method. Our contribution in this paper is threefold. First, we propose a PGM that encapsulates the relational learning problem. Second, we formulate various relational processing tasks as performing inference and learning in the PGM. Third, we propose an efficient and effective learning algorithm that can be trained end-to-end and unsupervised.

## 2 PROBLEM DEFINITION

We begin with formulating the relational learning problem as a machine learning problem: we observed a paired dataset $\mathbf{X} = \{ (\mathbf{a}^{(i)}, \mathbf{b}^{(i)}) \mid i \in [1..N] \}$ consisting of $N$ $i.i.d$ samples generated

from a joint distribution $p(\mathbf{a} \in \mathcal{A}, \mathbf{b} \in \mathcal{B})$; our goal is to learn a relational property between $\mathbf{a}^{(i)}$ and $\mathbf{b}^{(i)}$ irrespective of their absolute property. Furthermore, we want the learning to be unsupervised, e.g., we do not require a labeled dataset, such as $(\mathbf{a}^{(i)}, \mathbf{b}^{(i)}, z^{(i)})$ where $z^{(i)}$ is a target variable indicating $(\mathbf{a}^{(i)}, \mathbf{b}^{(i)})$'s relational property, nor do we require training examples that have the same (or different) relational property. There are two distinct features that separate our problem formulation from other unsupervised learning problem formulations:

1. We dissect the information in $\mathbf{X}$ into *relational property* and *absolute property*; relational property characterizes the relationship between $\mathbf{a}^{(i)}$ and $\mathbf{b}^{(i)}$, whereas absolute property represents specific features that independently describe $\mathbf{a}^{(i)}$ and $\mathbf{b}^{(i)}$.

2. Our goal is to learn a relational property among $\mathbf{X}$ *irrespective* of its absolute property, i.e., we want to learn a relational property that is *decoupled* from the absolute property.

In addition, we are interested in two related relational processing tasks: *relational discrimination*[1] and *relational mapping*[2] (VandenBos & APA, 2007).Relational discrimination allows us to differentiate $(\mathbf{a}^{(i)}, \mathbf{b}^{(i)})$ from $(\mathbf{a}^{(j)}, \mathbf{b}^{(j)})$ based on their relational properties, while relational mapping allows us to apply the relational property of $(\mathbf{a}^{(i)}, \mathbf{b}^{(i)})$ to a different set of data, for example, deduce that $\mathbf{b}^{(j)}$ is related to $\mathbf{a}^{(j)}$ in the same way that $\mathbf{b}^{(i)}$ is related to $\mathbf{a}^{(i)}$.

## 3 METHOD

Here we introduce the proposed VRL method for addressing the relational learning problem and discuss various optimization challenges unique to VRL.

### 3.1 VARIATIONAL RELATION LEARNING

The proposed VRL method consists of two parts: first, we encapsulate the relational learning problem with a PGM, called VRL-PGM; we then formulate various relational processing tasks as performing inference and learning in VRL-PGM.

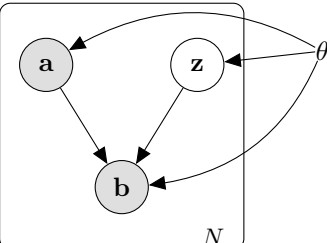

Figure 1: VRL-PGM: a probabilistic graphical model for representing the relational learning problem; the observed random variables $\mathbf{a}$ and $\mathbf{b}$ are generated from some random process (parameterized by $\theta$) involving an unobserved random variable $\mathbf{z}$.

The VRL-PGM model, shown in Fig. 1, generates data $\mathbf{a}$, $\mathbf{z}$, and $\mathbf{b}$ by sampling from PDFs that come from parametric families of distributions— $p_\theta(\mathbf{a})$, $p_\theta(\mathbf{z})$, $p_\theta(\mathbf{b}|\mathbf{a}, \mathbf{z})$—that are differentiable almost everywhere with respect to (w.r.t.) $\mathbf{a}$, $\mathbf{z}$, and $\theta$. In practice, we observe only a set of independent realizations $\{(\mathbf{a}^{(i)}, \mathbf{b}^{(i)}) \mid i \in [1..N]\}$ while the true parameter $\theta^*$ and the corresponding latent variables $\mathbf{z}^{(i)}$ are unobserved. A well-known property of the PGM shown in Fig. 1 is that random variables $\mathbf{a}$ and $\mathbf{z}$ are *independent* with no variables observed, but *not conditionally independent* when $\mathbf{b}$ is observed, i.e., $p_\theta(\mathbf{a}, \mathbf{z}) = p_\theta(\mathbf{a})p_\theta(\mathbf{z})$, $p_\theta(\mathbf{a}, \mathbf{z} \mid \mathbf{b}) \neq p_\theta(\mathbf{a} \mid \mathbf{b})p_\theta(\mathbf{z} \mid \mathbf{b})$ (Bishop, 2006). In VRL-PGM, the *absolute property* can be interpreted as representing the dependency between $\mathbf{a}$ and

---

[1]**Definition** (**Relational discrimination** in condition). *A discrimination based on the relationship between or among stimuli rather than on absolute features of the stimuli.*

[2]**Definition** (**Relational mapping**) *The ability to apply what one knows about one set of elements to a different set of elements.*

**b**, i.e., features in **a** that can be used to predict **b**, while the *relational property*, represented by the latent variable **z**, can be interpreted as any additional information not found in **a** but can help to better predict **b**. The key requirement that the learned relational property be decoupled from the absolute property is enforced by VRL-PGM's construction where **z**, which represents the relational property, and **a**, from which the absolute property is derived, are independent. The proposed VRL-PGM reflects our priority and compromise for using a PGM to represent the abstract relational learning problem: we sacrifice some identifiability of the original abstract problem (e.g., VRL-PGM artificailly introduces causal relationship between **a**, **b** and **z**, **b**) but we gained a rigorous and mathematical tractable PGM while achieving our primary objective of learning a decoupled (independent) relational property. Additional discussions on the connection between VRL-PGM and the relational learning problem is provided in appendix D.1.

Having established VRL-PGM, our primary learning objective is to approximate the unknown true likelihood function $p_\theta(\mathbf{b} \mid \mathbf{a}, \mathbf{z})$ and posterior $p_\theta(\mathbf{z} \mid \mathbf{a}, \mathbf{b})$ (note that by observing **b**, random variables **z** and **a** are no longer independent). Learning $p_\theta(\mathbf{z} \mid \mathbf{a}, \mathbf{b})$ provides us a way to infer $(\mathbf{a}^{(i)}, \mathbf{b}^{(i)})$'s relational property $\mathbf{z}^{(i)}$; moreover, it serves as a basis for performing *relational discrimination* where we compare relational properties between different pairs of data. Learning $p_\theta(\mathbf{b} \mid \mathbf{a}, \mathbf{z})$ allows us to perform *relational mapping* where we use the relational property of $(\mathbf{a}^{(i)}, \mathbf{b}^{(i)})$ to map from $\mathbf{a}^{(j)}$ to $\mathbf{b}^{(j)}$, i.e., $\mathbf{b}^{(j)} \sim p_\theta(\mathbf{b} \mid \mathbf{a}^{(j)}, \mathbf{z}^{(i)})$ where $\mathbf{z}^{(i)} \sim p_\theta(\mathbf{z} \mid \mathbf{a}^{(i)}, \mathbf{b}^{(i)})$.

## 3.2 Variational lower bound

We estimate the parameter for $p_\theta(\mathbf{b} \mid \mathbf{a}, \mathbf{z})$ by following the maximum-likelihood (ML) principle, and approximate the true posterior $p_\theta(\mathbf{z} \mid \mathbf{a}, \mathbf{b})$ with variational Bayesian approach. More specifically, we use a variational distribution $q_\phi(\mathbf{z} \mid \mathbf{a}, \mathbf{b})$, parameterized by $\phi$, to approximate the unknown (and often intractable) true posterior through maximizing a variational lower bound (Bishop, 2006). To derive such a lower bound, we first write the log-evidence as $\log p_\theta(\mathbf{X}) = \log p_\theta(\{(\mathbf{a}^{(i)}, \mathbf{b}^{(i)}) \mid i \in [1..N]\}) = \sum_{i=1}^N \log p_\theta(\mathbf{a}^{(i)}, \mathbf{b}^{(i)})$, where each term in the summation can be expressed as:

$$\log p_\theta(\mathbf{a}^{(i)}, \mathbf{b}^{(i)}) = D_{\mathrm{KL}}\Big(q_\phi(\mathbf{z} \mid \mathbf{a}^{(i)}, \mathbf{b}^{(i)}) \,\Big\|\, p_\theta(\mathbf{z} \mid \mathbf{a}^{(i)}, \mathbf{b}^{(i)})\Big)$$
$$+ \mathbb{E}_{q_\phi(\mathbf{z} \mid \mathbf{a}^{(i)}, \mathbf{b}^{(i)})}\Big[\log p_\theta(\mathbf{z}, \mathbf{a}^{(i)}, \mathbf{b}^{(i)}) - \log q_\phi(\mathbf{z} \mid \mathbf{a}^{(i)}, \mathbf{b}^{(i)})\Big]. \quad (1)$$

The first term on the RHS is the KL-divergence from $p_\theta(\mathbf{z} \mid \mathbf{a}^{(i)}, \mathbf{b}^{(i)})$ to $q_\phi(\mathbf{z} \mid \mathbf{a}^{(i)}, \mathbf{b}^{(i)})$, which provides a measure of dissimilarity between the two distributions; the second term on the RHS continues as:

$$\mathbb{E}_{q_\phi(\mathbf{z} \mid \mathbf{a}^{(i)}, \mathbf{b}^{(i)})}\Big[\log p_\theta(\mathbf{z}, \mathbf{a}^{(i)}, \mathbf{b}^{(i)}) - \log q_\phi(\mathbf{z} \mid \mathbf{a}^{(i)}, \mathbf{b}^{(i)})\Big]$$
$$= \mathbb{E}_{q_\phi(\mathbf{z} \mid \mathbf{a}^{(i)}, \mathbf{b}^{(i)})}\Big[\log p_\theta(\mathbf{b}^{(i)} \mid \mathbf{a}^{(i)}, \mathbf{z}) p_\theta(\mathbf{z}) p_\theta(\mathbf{a}^{(i)}) - \log q_\phi(\mathbf{z} \mid \mathbf{a}^{(i)}, \mathbf{b}^{(i)})\Big]$$
$$= \mathbb{E}_{q_\phi(\mathbf{z} \mid \mathbf{a}^{(i)}, \mathbf{b}^{(i)})}\Big[\log p_\theta(\mathbf{b}^{(i)} \mid \mathbf{a}^{(i)}, \mathbf{z}) + \log p_\theta(\mathbf{z}) - \log q_\phi(\mathbf{z} \mid \mathbf{a}^{(i)}, \mathbf{b}^{(i)})\Big] + \log p_\theta(\mathbf{a}^{(i)}), \quad (2)$$

where in the second line we use the fact that random variables **a** and **z** are independent. Substitute Eq. (2) back in (1) and rearrange terms gives us:

$$\log p_\theta(\mathbf{b}^{(i)} \mid \mathbf{a}^{(i)}) = D_{\mathrm{KL}}\Big(q_\phi(\mathbf{z} \mid \mathbf{a}^{(i)}, \mathbf{b}^{(i)}) \,\Big\|\, p_\theta(\mathbf{z} \mid \mathbf{a}^{(i)}, \mathbf{b}^{(i)})\Big) + \mathcal{L}(\theta, \phi; \mathbf{a}^{(i)}, \mathbf{b}^{(i)}) \quad (3)$$

where

$$\mathcal{L}(\theta, \phi; \mathbf{a}^{(i)}, \mathbf{b}^{(i)}) = \mathbb{E}_{q_\phi(\mathbf{z} \mid \mathbf{a}^{(i)}, \mathbf{b}^{(i)})}\Big[\log p_\theta(\mathbf{b}^{(i)} \mid \mathbf{a}^{(i)}, \mathbf{z}) + \log p_\theta(\mathbf{z}) - \log q_\phi(\mathbf{z} \mid \mathbf{a}^{(i)}, \mathbf{b}^{(i)})\Big]. \quad (4)$$

Since KL-divergence is non-negative, $\mathcal{L}(\theta, \phi; \mathbf{a}^{(i)}, \mathbf{b}^{(i)})$ (abbreviated as $\mathcal{L}^{(i)}$ for notation compactness) serves as a lower bound for the conditional log-likelihood $\log p_\theta(\mathbf{b}^{(i)} \mid \mathbf{a}^{(i)})$. Maximizing $\mathcal{L}^{(i)}$ w.r.t. $\phi$ and $\theta$ gives us both a ML estimate for $p_\theta(\mathbf{b} \mid \mathbf{a}, \mathbf{z})$ (by maximizing the first term inside the expectation in Eq. (4)) and a lower KL-divergence (the better $q_\phi(\mathbf{z} \mid \mathbf{a}^{(i)}, \mathbf{b}^{(i)})$ approximates the true posterior $p_\theta(\mathbf{z} \mid \mathbf{a}^{(i)}, \mathbf{b}^{(i)})$) as the conditional log-likelihood $\log p_\theta(\mathbf{b}^{(i)} \mid \mathbf{a}^{(i)})$ does not depend on $\phi$. The lower bound $\mathcal{L}^{(i)}$ can be maximized with gradient ascend methods; however, its gradients w.r.t. $\phi$ is difficult to obtain: the expectation in Eq. (4) is taken w.r.t. the distribution $q_\phi(\mathbf{z} \mid \mathbf{a}^{(i)}, \mathbf{b}^{(i)})$, which

is a function of $\phi$ (Paisley et al., 2012). To obtain efficient estimators for both $\mathcal{L}^{(i)}$ and its gradients, we adopt the *reparameterization trick* developed in Kingma & Welling (2014) where the random variable $\mathbf{z}$ is expressed as a transformation of another random variable $\boldsymbol{\epsilon} \sim p(\boldsymbol{\epsilon})$ that is independent of $\mathbf{a}$, $\mathbf{b}$, and $\phi$: $\mathbf{z} = g(\boldsymbol{\epsilon}, \mathbf{a}^{(i)}, \mathbf{b}^{(i)}, \phi)$ where $g$ is some differentiable and invertible transformation. Given such a change of variable, the lower bound $\mathcal{L}^{(i)}$ can be rewritten as:

$$\mathcal{L}^{(i)} = \mathbb{E}_{p(\boldsymbol{\epsilon})}\Big[\log p_\theta(\mathbf{b}^{(i)} \mid \mathbf{a}^{(i)}, \mathbf{z}) + \log p_\theta(\mathbf{z}) - \log q_\phi(\mathbf{z} \mid \mathbf{a}^{(i)}, \mathbf{b}^{(i)})\Big], \qquad (5)$$

where $\mathbf{z} = g(\boldsymbol{\epsilon}, \mathbf{a}^{(i)}, \mathbf{b}^{(i)}, \phi)$ and $\boldsymbol{\epsilon} \sim p(\boldsymbol{\epsilon})$. Note that the expectation in Eq. (5) is taken w.r.t. $p(\boldsymbol{\epsilon})$ and we can now approximate $\mathcal{L}^{(i)}$ with a Monte Carlo estimator:

$$\widetilde{\mathcal{L}}^{(i)} = \frac{1}{L}\sum_{l=1}^{L}\log p_\theta(\mathbf{b}^{(i)} \mid \mathbf{a}^{(i)}, \mathbf{z}^{(i,l)}) + \log p_\theta(\mathbf{z}^{(i,l)}) - \log q_\phi(\mathbf{z}^{(i,l)} \mid \mathbf{a}^{(i)}, \mathbf{b}^{(i)}), \qquad (6)$$

where $\mathbf{z}^{(i,l)} = g(\boldsymbol{\epsilon}^{(i,l)}, \mathbf{a}^{(i)}, \mathbf{b}^{(i)}, \phi)$ and $\boldsymbol{\epsilon}^{(i,l)} \sim p(\boldsymbol{\epsilon})$. The lower bound for a minibatches of data $\mathbf{X}^M = \{(\mathbf{a}^{(i)}, \mathbf{b}^{(i)}) \mid i \in [1..M]\}$ can be approximated by $\widetilde{\mathcal{L}}(\theta, \phi; \mathbf{X}^M) = \frac{N}{M}\sum_{i=1}^{M}\widetilde{\mathcal{L}}^{(i)}$. And finally, the gradients $\nabla_{\theta,\phi}\widetilde{\mathcal{L}}(\theta, \phi; \mathbf{X}^M) = \frac{N}{M}\sum_{i=1}^{M}\nabla_{\theta,\phi}\widetilde{\mathcal{L}}^{(i)}$ can be computed in a straightforward manner and used to update the parameters $\theta$ and $\phi$ with stochastic optimization methods, such as SGD. Finally, additional discussion is provided in appendix D.2 where we explain how the optimization of the variational lower bound in Eq. 4 naturally encourages the indpendence of $\mathbf{z}$ and $\mathbf{a}$; we also discuss possible extensions to Eq. 4 to explicitly safeguard against introducing dependency between $\mathbf{z}$ and $\mathbf{a}$.

### 3.3 OPTIMIZATION CHALLENGES

The proposed VRL method introduces unique challenges to the variational lower bound optimization problem (see Sønderby et al. (2016) and Bowman et al. (2016) for other known challenges). To explain these challenges, we first break down VRL's parameter updating process into the following steps (using a single datapoint as an example): (1) a datapoint $(\mathbf{a}^{(i)}, \mathbf{b}^{(i)})$ is selected; (2) sample $\mathbf{z}^{(i)} \sim q_{\phi_k}(\mathbf{z} \mid \mathbf{a}^{(i)}, \mathbf{b}^{(i)})$ by using the current parameter $\phi_k$; (3) evaluate $\widetilde{\mathcal{L}}^{(i)}$ by using $\phi_k, \theta_k$; (4) calculate gradients $g = \nabla_{\theta_k, \phi_k}\widetilde{\mathcal{L}}^{(i)}$; (5) use gradients $g$ to update $\phi_k, \theta_k$ and get new parameters $\phi_{k+1}, \theta_{k+1}$. This parameter updating process can be depicted with an information flow diagram shown in Fig. 2a. Ideally, we would like every path in Fig. 2a to contribute to the evaluation of all the terms in its reachable nodes in order to obtain meaningful gradients for updating its associated parameters; however, there are two situations where this is not the case. The first situation, called

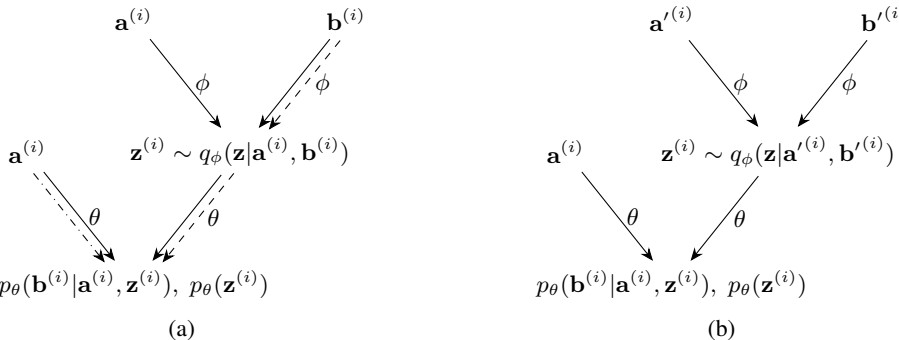

Figure 2: Information flow diagrams depicting VRL's parameter updating process, where each path uses its associated parameters to propagate information in the forward direction and gradients in the backward direction: (a) Unmodified parameter updating process, where overfitting occurs when the learning of $p_\theta(\mathbf{b}^{(i)} \mid \mathbf{a}^{(i)}, \mathbf{z}^{(i)})$ rely only on the dash-dotted path (*deterministic-mapping*) or the dashed path (*information-shortcut*); (b) Parameter updating process improved with RPDA.

*information-shortcut*, occurs when the learning of $p_\theta(\mathbf{b}^{(i)} \mid \mathbf{a}^{(i)}, \mathbf{z}^{(i)})$ rely entirely on the dashed path in Fig. 2a; more specifically, the dashed (shortcut) path directly propagates $\mathbf{b}^{(i)}$ through $\mathbf{z}^{(i)}$

to $p_\theta(\mathbf{b}^{(i)} \mid \mathbf{a}^{(i)}, \mathbf{z}^{(i)})$ and, as a result, the relational property $\mathbf{z}^{(i)}$ may learn only to encode the absolute property of $\mathbf{b}^{(i)}$. The second situation, called *deterministic-mapping*, occurs when $\mathbf{b}^{(i)}$ can be fully characterized by $\mathbf{a}^{(i)}$; in this case, the learning of $p_\theta(\mathbf{b}^{(i)} \mid \mathbf{a}^{(i)}, \mathbf{z}^{(i)})$ may rely only on the dash-dotted path in Fig. 2a. While both situations can be viewed as a overfitting problem, *deterministic-mapping* says more about the data itself and exposes a potential limitation of VRL: a decoupled relational property is primarily learned through exploring information not found in $\mathbf{a}$ but can help to better predict $\mathbf{b}$, and when $\mathbf{a}$ fully characterizes $\mathbf{b}$, VRL no longer need to explore information beyond $\mathbf{a}$ to predict $\mathbf{b}$ and this may prevent VRL from learning a meaningful relational property. On the other hand, *information-shortcut* is caused by short-cutting the parameter updating process, which we may overcome with additional regularization techniques.

Here we propose two approaches for mitigating the *information-shortcut* problem by disrupting the flow of information passing through the shortcut path. In the first approach, we restrict the flow of information by constraining the expressiveness of the latent variable $\mathbf{z}$; for example, by adopting an informative prior with restrictive constraint, such as $p_\theta(\mathbf{z}) = \mathcal{N}(\mathbf{z}; 0, \sigma^2 \mathbf{I})$, $\sigma \ll 1$, or representing $\mathbf{z}$ with a discrete random variable (assuming we know a priori the underlying relational proproty are discrete). In the second approach, we propose a novel data augmentation strategy—relation-preserving data augmentation (RPDA)—that aims to eliminate the shortcut path. First, we define a set of relation preserving functions $D = \{ d(\mathbf{a}, \mathbf{b}; r) \mid r \in R (\text{some index set}),\ d: \mathcal{A} \times \mathcal{B} \to \mathcal{A} \times \mathcal{B} \}$ where the data relationship is preserved in the following sense: $p_\theta(\mathbf{z} \mid \mathbf{a}, \mathbf{b}) = p_\theta(\mathbf{z} \mid \mathbf{a}', \mathbf{b}')$ for $(\mathbf{a}', \mathbf{b}') = d(\mathbf{a}, \mathbf{b}; r), \forall r$. Assuming we have access to such a $D$, the proposed RPDA strategy then seek to optimize a modified lower bound estimator $\widetilde{\mathcal{L}}_{\text{RPDA}}^{(i)}$:

$$\widetilde{\mathcal{L}}_{\text{RPDA}}^{(i)} = \frac{1}{L} \sum_{l=1}^{L} \log p_\theta(\mathbf{b}^{(i)} \mid \mathbf{a}^{(i)}, \mathbf{z}^{(i,l)}) + \log p_\theta(\mathbf{z}^{(i,l)}) - \log q_\phi(\mathbf{z}^{(i,l)} \mid \mathbf{a}'^{(i)}, \mathbf{b}'^{(i)}), \quad (7)$$

$$\text{where} \quad (\mathbf{a}'^{(i)}, \mathbf{b}'^{(i)}) = d(\mathbf{a}^{(i)}, \mathbf{b}^{(i)}; r^{(i)}),\ \ r^{(i)} \sim \mathcal{U}(R),$$

and $\mathbf{z}^{(i,l)} = g(\boldsymbol{\epsilon}^{(i,l)}, \mathbf{a}'^{(i)}, \mathbf{b}'^{(i)}, \phi)$, $\boldsymbol{\epsilon}^{(i,l)} \sim p(\boldsymbol{\epsilon})$. Note that due to the relation preserving property of $D$, we have $q_\phi(\mathbf{z}^{(i,l)} \mid \mathbf{a}'^{(i)}, \mathbf{b}'^{(i)}) = q_\phi(\mathbf{z}^{(i,l)} \mid \mathbf{a}^{(i)}, \mathbf{b}^{(i)})$ and, therefore, $\widetilde{\mathcal{L}}_{\text{RPDA}}^{(i)}$ is equivalent to $\widetilde{\mathcal{L}}^{(i)}$ in Eq. (6). When we optimize with $\widetilde{\mathcal{L}}_{\text{RPDA}}^{(i)}$, the parameter updating process can be redrawn in Fig. 2b, where now the learning of $p_\theta(\mathbf{b}^{(i)} \mid \mathbf{a}^{(i)}, \mathbf{z}^{(i)})$ can no longer rely solely on the shortcut path to propagate $\mathbf{b}'^{(i)}$ since it differs from $\mathbf{b}^{(i)}$ by a non-deterministic factor $r^{(i)}$. In practice, it may seem unrealistic to assume that we can construct a set of RPDA functions $D$ without extensive knowledge of the underlying relational property. However, we can treat data augmentation as a form of regularization and construct a $D$ that reflects our prior knowledge and belief of the underlying system (Ronneberger et al., 2015; Perez & Wang, 2017). For example, if we want the learning to be rotation invariance (a common theme in computer vision applications), we can construct a $D$ that consists of image rotation augmentations, e.g., $d(\mathbf{a}, \mathbf{b}; r) = (\text{rot}(\mathbf{a}, r), \text{rot}(\mathbf{b}, r))$ where $\text{rot}(\mathbf{x}, r)$ rotates the image $\mathbf{x}$ by $r \in R = [0, 360)$ degrees (note that both $\mathbf{a}$ and $\mathbf{b}$ are rotated by the same amount). Additional remarks on the practical applicability of RPDA is provided in appendix D.3, and a detailed ablation study is provided in appendix C.

To summarize this section, the proposed VRL method with RPDA is described in Algorithm 1.

## 4  RELATED WORK

Machine learning approaches for relational processing have gained increasing interest and attention in the literature. Most of these methods focus on high-level cognitive tasks, such as visual Q&A and state prediction for complex-physics systems, and derive their relational processing capabilities from learning with clever designed neural networks (Hill et al., 2019; Santoro et al., 2017; Raposo et al., 2017; Battaglia et al., 2018; 2016; Wu et al., 2015; Reed et al., 2015; van Steenkiste et al., 2018; Chang et al., 2016; Fragkiadaki et al., 2015). Our work differ from these methods in two ways: (1) our primary focus is addressing the relational learning problem where we want to learn a decoupled relational property; (2) we enforce the decoupling requirement on the learned relational property with a PGM, which gives us the flexibility to use any compatible inference method or function approximation and still satisfy the decoupling requirement.

---

**Algorithm 1** VRL with RPDA

---

**procedure** VRL($\mathbf{X}$, $p(\boldsymbol{\epsilon})$, $D$)       ▷ If RPDA not available, $D = \{\, \text{id}(\cdot) \mid (\mathbf{a}, \mathbf{b}) = \text{id}(\mathbf{a}, \mathbf{b}) \,\}$
    Initialize parameters $\theta, \phi$
    **while** not convergence of parameters $(\theta, \phi)$ **do**
        Sample minibatch $\mathbf{X}^M = \{\, (\mathbf{a}^{(i)}, \mathbf{b}^{(i)}) \mid i \in [1..M] \,\}$ from $\mathbf{X}$.
        Sample $\boldsymbol{\epsilon}^{(i,l)} \sim p(\boldsymbol{\epsilon})$, $r^{(i)} \sim \mathcal{U}(R)$, $i = 1, ..., M$, $l = 1, ..., L$.
        Run RPDA and obtain $(\mathbf{a}'^{(i)}, \mathbf{b}'^{(i)}) = d(\mathbf{a}^{(i)}, \mathbf{b}^{(i)}; r^{(i)})$, $i = 1, ..., M$.
        Compute gradients $g = \nabla_{\theta,\phi} \widetilde{\mathcal{L}}\left(\theta, \phi; \mathbf{X}^M\right) = \frac{N}{M} \sum_{i=1}^{M} \nabla_{\theta,\phi} \widetilde{\mathcal{L}}_{\text{RPDA}}^{(i)}$ (see Eq. (7)).
        Update parameters $\theta, \phi$ using gradients $g$ (e.g. SGD).
    **end while**
    **return** $\theta, \phi$
**end procedure**

---

Conventional unsupervised learning methods can also be appied to our problem setting (Kingma & Welling, 2014; Goodfellow et al., 2014; Mikolov et al., 2013a;b; Song et al., 2007); however, these methods learn a single representation for the data with superimposing information about their relational and absolute property. The difficulty of decoupling the relational property from the learned representation constitute a major obstacle to relational reasoning.

Other related work include methods on learning a disentangled representations with applications in style-transfer, image-to-image translation, domain adaptation, etc. (Huang et al., 2018; Chen et al., 2016; Tenenbaum & Freeman, 1997; Higgins et al., 2017; Bousmalis et al., 2016; Mathieu et al., 2016; Tulyakov et al., 2017; Denton & Birodkar, 2017; Villegas et al., 2017; Donahue et al., 2017; Shen et al., 2017) Most of these methods strive to learn a disentangled representations of *content* and *style* (or *pose* for video sequence data) where *content* is generically defined as the underling spatial structure, and *style* as the rendering of the structure. In comparison, our work can be viewed as learning a disentangled representations of *relational* and *absolute property*; however, we argue that *style-content* separation is fundamentally different from *relational-absolute* separation. More specifically, we consider both *style* and *content* information as *absolute property* (both describe features of an individual data), while *relational property* provides new information on the (abstract) relationship between the paired data.

## 5 EXPERIMENT

In this section, we present experimental results from applying the proposed VRL method to a set of relational learning tasks designed with the MNIST dataset (LeCun & Cortes, 2010).

### 5.1 MNIST RELATIONAL LEARNING TASK

To setup a relational learning task, a paired dataset $\mathbf{X} = \{\, (\mathbf{a}^{(i)}, \mathbf{b}^{(i)}) \mid i \in [1..N] \,\}$ was generated by the following steps: (1) the MNIST dataset $\{\, (\mathbf{x}^{(i)}, y^{(i)}) \mid i \in [1..T] \,\}$, where $\mathbf{x}^{(i)}$ and $y^{(i)}$ are the digit images and their labels, was augmented with applying five evenly-spaced rotations to each of the image $\mathbf{x}^{(i)}$ to get $\{\, (\mathbf{x}^{(i,j)}, y^{(i)}) \mid i \in [1..T], j \in [1..5] \,\}$ (examples of augmented images are shown in Fig. 3); (2) individual datapoint $(\mathbf{a}^{(i)}, \mathbf{b}^{(i)})$ of $\mathbf{X}$ was chosen to be randomly rotated images of $\mathbf{x}^{(i)}$, i.e., $\mathbf{X} = \{\, (\mathbf{a}^{(i)} = \mathbf{x}^{(j,k)}, \mathbf{b}^{(i)} = \mathbf{x}^{(j,l)}) \mid j \sim \mathcal{U}([1..T]),\ k,l \sim \mathcal{U}([1..5]) \,\}$. Note that

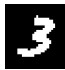 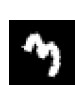 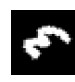 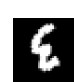 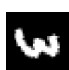

Figure 3: Examples of a MNIST digit augmented with five evenly-spaced rotations (from left to right: $\mathbf{x}^{(i,1)} : 0°$, $\mathbf{x}^{(i,2)} : 72°$, $\mathbf{x}^{(i,3)} : 144°$, $\mathbf{x}^{(i,4)} : 216°$, $\mathbf{x}^{(i,5)} : 288°$).

there are five uniquely defined rotational relationships between $(\mathbf{a}^{(i)}, \mathbf{b}^{(i)})$ in $\mathbf{X}$ and since they are randomly selected, the rotational relationship (relational property) is decoupled from the absolute properties of $(\mathbf{a}^{(i)}, \mathbf{b}^{(i)})$. Here, we use $\mathbf{X}$ to assess VRL's capability of discovering the underlying

relational property (rotational relationships) irrespective of their absolute property. Additional MNIST relational learning tasks are introduces in appendix B.1.

The MNIST relational learning experiment presented above may seem contrived at first glance but, upon deeper examination, represents a novel and unique problem setting that exemplifies a key relational learning challenge—learning a decoupled relational property—for existing unsupervised learning methods (see Sec. 4 for related work). We argue that *any* unsupervised method that can successfully solve the above problem (or any relational learning problem) must, at a minimum, simultaneously accomplish the following two goals:

1. An unsupervised learning mechanism that captures and preserves the data relationships (relational property), e.g, capturing and preserving the rotational relationship between $(\mathbf{a}^{(i)}, \mathbf{b}^{(i)})$ during learning.

2. An unsupervised learning mechanism that decouples absolute property from the learned data relationships, e.g., learning a rotational relationship between $(\mathbf{a}^{(i)}, \mathbf{b}^{(i)})$ that does not depend on the absolute property (digit representation, rotation, etc.) of individual $\mathbf{a}^{(i)}, \mathbf{b}^{(i)}$.

While most existing unsupervised methods are well-equipped to accomplish the first goal, we argue that the second goal presents itself as a major challenge; more specifically, to the best of our knowledge, most existing unsupervised methods focus on modeling all aspect of $(\mathbf{a}^{(i)}, \mathbf{b}^{(i)})$ and learn data relationships either jointly with the absolute properties of $\mathbf{a}^{(i)}, \mathbf{b}^{(i)}$ or as their derivatives (including graph-based methods that use edges to represent relationships). For such methodology, the learned data relationships necessarily entangle/couple with the absolute properties of the data, and therefore, the fundamental relational learning challenge—decoupling relational property—is still unresolved. A key insight into why the proposed VRL framework is capable of overcoming this challenge is recognizing that VRL takes a more targeted approach and learns data relationships as a stand-alone entity (represented by the latent variable $\mathbf{z}$) that is designed, through the construction of VRL-PGM, to be independent of $\mathbf{a}$ (however, we note that in VRL the learned relational property $\mathbf{z}$ is *only* independent of $\mathbf{a}$ but not $\mathbf{b}$; a discussion of this compromising fact and its implications is provided in appendix D.1).

## 5.2 Implementation

For these experiments, we adopted a two-dimensional latent variable $\mathbf{z} \in \mathbb{R}^2$ and let the prior $p_\theta(\mathbf{z})$ be the bivariate normal distribution. We let $p_\theta(\mathbf{b} \mid \mathbf{a}^{(i)}, \mathbf{z}^{(i,l)})$ be a multivariate Bernoulli distribution whose probability parameters are computed from a given $\mathbf{a}^{(i)}$ and $\mathbf{z}^{(i,l)}$ with an autoencoder-like neural network $f_\theta^{\text{dec}}\left(f_\theta^{\text{enc}}(\mathbf{a}^{(i)}), \mathbf{z}^{(i,l)}\right)$. We let the approximated posterior $q_\phi(\mathbf{z} \mid \mathbf{a}'^{(i)}, \mathbf{b}'^{(i)})$ be a bivariate Gaussian distribution with a diagonal covariance $\mathcal{N}(\mathbf{z}; \boldsymbol{\mu}^{(i)}, (\boldsymbol{\sigma}^{(i)})^2 \mathbf{I})$ where $\boldsymbol{\mu}^{(i)}$ and $\boldsymbol{\sigma}^{(i)}$ are the output of a neural network $f_\phi^q(\mathbf{a}'^{(i)}, \mathbf{b}'^{(i)})$. For RPDA, we constructed a $D$ that consists of image rotation augmentations: $D = \{ (\text{rot}(\mathbf{a}, r), \text{rot}(\mathbf{b}, r)) \mid r \in [0, 360) \}$. Detailed experimental setup is described in appendix A.

## 5.3 Results

**Relational discrimination.** We trained VRL on X and used the approximated posterior $q_\phi(\mathbf{z} \mid \mathbf{a}, \mathbf{b})$ to infer the relational property of a hold-out dataset. Figure 4a shows a scatter plot of the relational property inferred by VRL where we can see that the approximated posterior accurately cluster(discriminate) data with the same(different) rotational relationship together(apart). Here, we compared VRL with variational autoencoder (VAE) (Kingma & Welling, 2014) and InfoGAN (Chen et al., 2016), both of which can learn data representations of X in a completely unsupervised manner. In the application of VAE, we adopted a 2-D latent space and use the encoder trained on X to inferred the latent varible of a hold-out dataset; the resulting scatter plot is shown in Fig. 4b where we see that VAE failed to discriminate data based on their relational property (rotational relationship). Next, InfoGAN has demonstrated its ability to learn disentangled representations (represented by structured latent codes $c_1, c_2, ..., c_L$) through generative modelling. Although inferring latent codes for a given data point is a non-trivial task for InfoGAN, we can examine the learned latent representation by manipulating the latent codes and visually inspect the generated

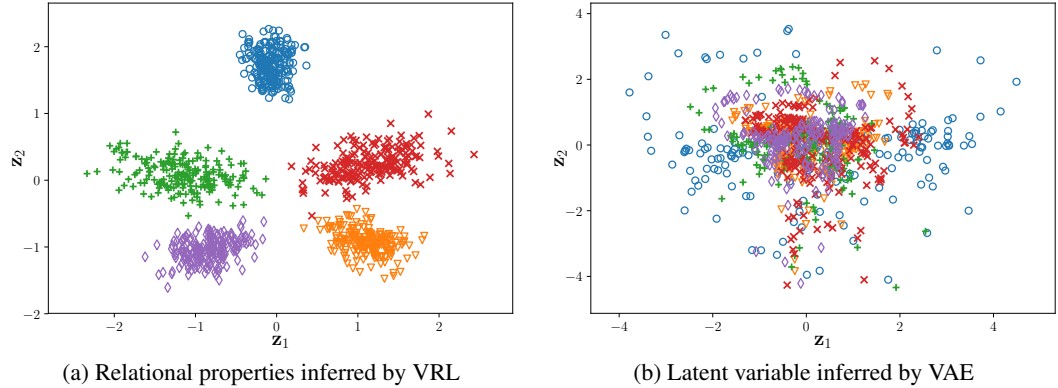

(a) Relational properties inferred by VRL                    (b) Latent variable inferred by VAE

Figure 4: Scatter plots of the 2-D latent variable (showing only the mean $\boldsymbol{\mu}$) of hold-out datasets inferred by VRL in (a) and VAE in (b) (relationship labels: $\bigcirc : 0°$, $\triangledown : 72°$, $+ : 144°$, $\times : 216°$, $\diamondsuit : 288°$).

random samples. We followed the examples in Chen et al. (2016) and modeled the latent codes with one categorical code $c_1 \sim \text{Cat}(K = 10, p = 0.1)$ (model discontinuous variation in data) and two continuous codes $c_2, c_3 \sim \text{Unif}(-1, 1)$ (capture continuous variations in style). Figure 5 shows examples of generated images from manipulating the latent codes; it is clear that none of the latent codes (or a combination of them) distinctively capture the full range of relational property (rotational relationship). Figures 4b and 5 illustrate a major challenge of using VAE and InfoGAN (and other related methods) for relational learning: these methods learn a single representation that encodes both relational and absolute property of X and it is difficult to dissect the relational property from the learned representation. Detailed experimental setup for both VAE and InfoGAN is described in appendix A.

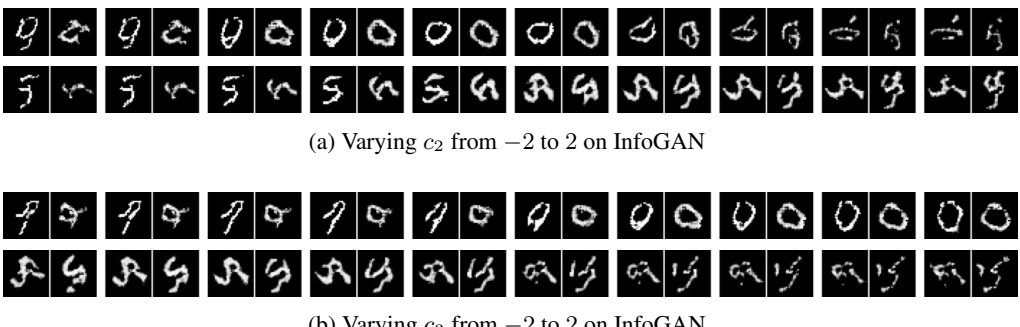

(a) Varying $c_2$ from $-2$ to 2 on InfoGAN

(b) Varying $c_3$ from $-2$ to 2 on InfoGAN

Figure 5: Manipulating latent codes of InfoGAN on MNIST where each row represents random samples from varying continuous latent code $c_2$ in (a) and $c_3$ in (b) while other latent codes and noise are fixed; different rows correspond to different categorical code $c_1$.

**Relational mapping.** We evaluated VRL's learned likelihood function $p_\theta(\mathbf{b} \mid \mathbf{a}, \mathbf{z})$ by visualizing the predicted images given $\mathbf{a}$ and $\mathbf{z}$. We chose $\mathbf{a}$ from a hold-out dataset and $\mathbf{z}$ from: (1) direct sampling in the latent space; (2) relational property inferred from a source datapoint $(\mathbf{a}_s, \mathbf{b}_s)$. Figure 6a shows predicted images with $\mathbf{z}$ sampled from the latent space shown in Fig. 4a. Figure 6b shows examples of relational mappings from $\mathbf{a}^{(c)}$ to $\mathbf{b}^{(r,c)}$ by applying the relational property inferred from a source datapoint $(\mathbf{a}_s, \mathbf{b}_s^{(r)})$. At first glance, the results in Fig. 6b resemble that of style-transfer, but they are fundamentally different: in style-transfer, the image $\mathbf{b}^{(r,c)}$ is generated by applying the *style* of $\mathbf{b}_s^{(r)}$ to the *content* of $\mathbf{a}^{(c)}$, whereas VRL generates image $\mathbf{b}^{(r,c)}$ by applying the *relationship* of $(\mathbf{a}_s, \mathbf{b}_s^{(r)})$ to the image $\mathbf{a}^{(c)}$. It is evident from Fig. 6b that predicted images $\mathbf{b}^{(r,c)}$ do not share similar *style* to $\mathbf{b}_s^{(r)}$, but rather the same rotational relationship w.r.t. $\mathbf{a}^{(c)}$ and $\mathbf{a}_s$.

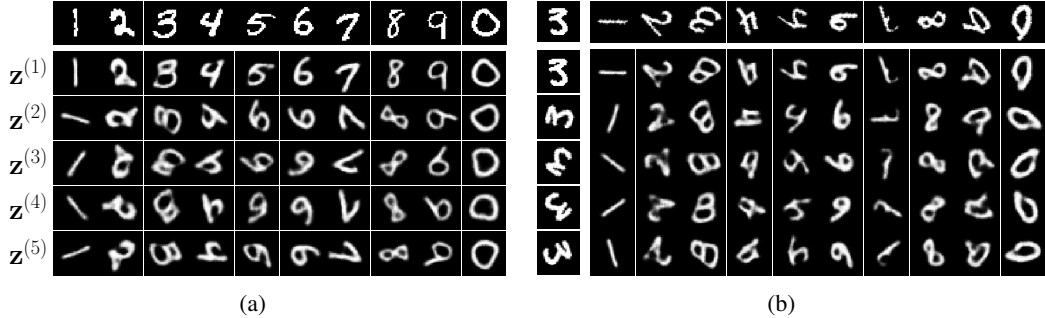

(a)                          (b)

Figure 6: Examples of images predicted by VRL: (a) images predicted from sampled latent variables (sampling the centroid of each cluster in Fig. 4a: "$\bigcirc$" $\rightarrow \mathbf{z}^{(1)}$, "$\nabla$" $\rightarrow \mathbf{z}^{(2)}$, "$+$" $\rightarrow \mathbf{z}^{(3)}$, "$\times$" $\rightarrow \mathbf{z}^{(4)}$, "$\diamond$" $\rightarrow \mathbf{z}^{(5)}$) and each image $\mathbf{b}^{(r,c)}, 1 \leq r \leq 5, 1 \leq c \leq 10$, was predicted from an image $\mathbf{a}^{(c)}$ (shown in the top row) and a pre-selected latent variable $\mathbf{z}^{(r)}$ using $\mathbf{b}^{(r,c)} \sim p_\theta(\mathbf{b} \mid \mathbf{a}^{(c)}, \mathbf{z}^{(r)})$; (b) examples of relational mappings of top row images by applying relationships inferred from pairs of source images $(\mathbf{a}_s, \mathbf{b}_s^{(r)})$ (shown in the left-most column with $\mathbf{a}_s, \mathbf{b}_s^{(1)}, ..., \mathbf{b}_s^{(5)}$ arranged from top to bottom) and each image $\mathbf{b}^{(r,c)}, 1 \leq r \leq 5, 1 \leq c \leq 10$ was generated by $\mathbf{b}^{(r,c)} \sim p_\theta(\mathbf{b} \mid \mathbf{a}^{(c)}, \mathbf{z}^{(r)})$ where $\mathbf{z}^{(r)} \sim q_\phi(\mathbf{z} \mid \mathbf{a}_s, \mathbf{b}_s^{(r)})$.

Additional experimental results and discussion are provided in appendix B and D.4, respectively.

## 6   DISCUSSION AND CONCLUSION

A core component of the proposed VRL method is approximating the intractable posterior with variational inference (VI) methods. There is a vast literature on the subject of VI that we can leverage to further improve and extend VRL; for example, prior works have proposed flexible and complex approximated posterior distributions that we can use to learn a rich posterior approximations for characterizing the relational property (Rezende & Mohamed, 2015; Dinh et al., 2014). Another interesting idea to explore is learning a generative model for VRL: in this work, the primary learning objective is maximizing a variational lower bound derived from VRL-PGM; however, with the advent of computationally efficient methods for learning a generative model, it would be interesting to include another learning objective that directly models the data generating aspect of VRL-PGM (Goodfellow et al., 2014; Larsen et al., 2015).

The proposed method comes with both advantages and disadvantages: the main advantage of VRL lies in its relational learning capabilities; however, this may also be one of its disadvantages. More specifically, VRL can learn a decoupled relational property even when it is coupled with the absolute property, i.e., VRL is oblivious to the coupling information between the two properties (an example with coupled relational property is provided in appendix B.1.1). Nevertheless, such information may be of interest to the user, and in this regard, VRL provides only a partial view of the data.

In conclusion, the proposed VRL method is an efficient and effective unsupervised learning method for addressing the relational learning problem where our goal is to learn a decoupled relational property. By dissecting the data information into decoupled *relational* and *absolute property*, we hope VRL can bring new insight into everyday data analysis and ultimately find applications for a wide variety of problems.

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

## A  EXPERIMENTAL SETUP

Recall that we adopted a two-dimensional latent variable $\mathbf{z} \in \mathbb{R}^2$ and let the prior $p_\theta(\mathbf{z})$ be the bivariate normal distribution. For binary valued data (e.g., MNIST dataset), we let the likelihood function $p_\theta(\mathbf{b} \in \mathcal{B} \mid \mathbf{a}^{(i)}, \mathbf{z}^{(i,l)})$ of VRL be a multivariate Bernoulli distribution whose probability parameters $\mathbf{p}^{(i,l)}$ are computed from a given $\mathbf{a}^{(i)}$ and $\mathbf{z}^{(i,l)}$ with an autoencoder-like neural network $f_\theta^{\text{dec}}\left(f_\theta^{\text{enc}}(\mathbf{a}^{(i)}), \mathbf{z}^{(i,l)}\right)$:

$$f_\theta^{\text{enc}} : \mathbf{a}^{(i)} \to \text{Conv(3x3x8)} \to \text{Conv(3x3x32)} \to \text{Conv(3x3x128)} \to \text{FC}(20) \to \mathbf{h}^{(i)} \in \mathbb{R}^{20}$$

$$f_\theta^{\text{dec}} : [\mathbf{h}^{(i)} \in \mathbb{R}^{20}, \mathbf{z}^{(i,l)} \in \mathbb{R}^2] \to \text{FC} \to \text{Conv}^T(\text{3x3x128}) \to \text{Conv}^T(\text{3x3x32}) \to \text{Conv}^T(\text{3x3x8})$$

$$\to \text{Conv(1x1x1)} \xrightarrow{\text{Sigmoid}} [0,1]^{\dim(\mathcal{B})},$$

where $\text{Conv}(\cdot)$ is a strided (stride 2) convolutional layer and $\text{Conv}^T(\cdot)$ is a transposed convolutional layer. We used batch-normalization after most layers and leaky rectified linear units (with leaky rate 0.01) as nonlinear activation function. We let the approximated posterior $q_\phi(\mathbf{z} \mid \mathbf{a}'^{(i)}, \mathbf{b}'^{(i)})$ of VRL be a bivariate Gaussian distribution with a diagonal covariance $\mathcal{N}(\mathbf{z}; \boldsymbol{\mu}^{(i)}, (\boldsymbol{\sigma}^{(i)})^2 \mathbf{I})$ where $\boldsymbol{\mu}^{(i)}$ and $\boldsymbol{\sigma}^{(i)}$ are the output of a neural network $f_\phi^q(\mathbf{a}'^{(i)}, \mathbf{b}'^{(i)})$:

$$f_\phi^q : [\mathbf{a}'^{(i)}, \mathbf{b}'^{(i)}] \to \text{Conv(3x3x8)} \to \text{Conv(3x3x32)} \to \text{Conv(3x3x128)} \to \text{FC}(4) \to [\boldsymbol{\mu}^{(i)}, \boldsymbol{\sigma}^{(i)}].$$

We sampled from the posterior $\mathbf{z}^{(i,l)} \sim \mathcal{N}(\mathbf{z}; \boldsymbol{\mu}^{(i)}, (\boldsymbol{\sigma}^{(i)})^2 \mathbf{I})$ using $\mathbf{z}^{(i,l)} = g(\boldsymbol{\epsilon}^{(i,l)}, \mathbf{a}'^{(i)}, \mathbf{b}'^{(i)}, \phi) = \boldsymbol{\mu}^{(i)} + \boldsymbol{\sigma}^{(i)} \odot \boldsymbol{\epsilon}^{(i,l)}$ where $\boldsymbol{\epsilon}^{(i,l)} \sim p(\boldsymbol{\epsilon}) = \mathcal{N}(\mathbf{0}, \mathbf{I})$ and set $L = 1$. We used image rotation augmentations: $D = \{(\text{rot}(\mathbf{a}, r), \text{rot}(\mathbf{b}, r)) \mid r \in [0, 360)\}$ for constructing RPDA. In this case, the learning objective $\widetilde{\mathcal{L}}_{\text{RPDA}}^{(i)}$ in Eq. (7) can further be derived as (see next section for the derivation):

$$\widetilde{\mathcal{L}}_{\text{RPDA}}^{(i)} = \frac{1}{2} \sum_{j=1}^2 \left( 1 + \log((\sigma_j^{(i)})^2) - (\mu_j^{(i)})^2 - (\sigma_j^{(i)})^2 \right)$$
$$+ \sum_k b_k^{(i)} \log p_k^{(i,1)} + (1 - b_k^{(i)}) \log(1 - p_k^{(i,1)}). \tag{8}$$

$$\text{where} \quad (\boldsymbol{\mu}^{(i)}, \boldsymbol{\sigma}^{(i)}) = f_\phi^q(\mathbf{a}'^{(i)}, \mathbf{b}'^{(i)}), \quad \mathbf{p}^{(i,l)} = f_\theta^{\text{dec}}\left(f_\theta^{\text{enc}}(\mathbf{a}^{(i)}), \mathbf{z}^{(i,1)}\right),$$
$$\mathbf{z}^{(i,1)} = \boldsymbol{\mu}^{(i)} + \boldsymbol{\sigma}^{(i)} \odot \boldsymbol{\epsilon}^{(i,1)}, \quad \boldsymbol{\epsilon}^{(i,1)} \sim \mathcal{N}(\mathbf{0}, \mathbf{I}),$$
$$(\mathbf{a}'^{(i)}, \mathbf{b}'^{(i)}) = (\text{rot}(\mathbf{a}^{(i)}, r^{(i)}), \text{rot}(\mathbf{b}^{(i)}, r^{(i)})), \quad r^{(i)} \sim \mathcal{U}([0, 360)).$$

Parameters $\theta$ and $\phi$ were jointly trained to maximize Eq. (8) using Adam optimizer (learning rate = 0.0001, $\beta1 = 0.9$, $\beta2 = 0.999$) (Kingma & Ba, 2014); minibatches of size $M = 100$ were used.

In our VAE implementation we used the following network architecture for encoder, $\text{Enc}(\mathbf{a}^{(i)}, \mathbf{b}^{(i)})$, and decoder, $\text{Dec}(\mathbf{z}^{(i,l)})$:

$$\text{Enc} : [\mathbf{a}^{(i)}, \mathbf{b}^{(i)}] \to \text{Conv(3x3x16)} \to \text{Conv(3x3x64)} \to \text{Conv(3x3x256)} \to \text{FC}(4)$$
$$\to [\boldsymbol{\mu}^{(i)} \in \mathbb{R}^2, \boldsymbol{\sigma}^{(i)} \in \mathbb{R}^2]$$
$$\text{Dec} : [\mathbf{z}^{(i,l)} \in \mathbb{R}^2] \to \text{FC} \to \text{Conv}^T(\text{3x3x256}) \to \text{Conv}^T(\text{3x3x64}) \to \text{Conv}^T(\text{3x3x16})$$
$$\to \text{Conv(1x1x1)} \xrightarrow{\text{Sigmoid}} [0,1]^{\dim(\mathcal{B})}.$$

We used Adam optimizer with the same hyperparameter setup as before.

For InfoGAN implementation, we followed the setup described in Chen et al. (2016) except that each datapoint consists a paired MNIST image $(\mathbf{a}^{(i)}, \mathbf{b}^{(i)})$.

## A.1 DERIVATION OF TRAINING OBJECTIVE FUNCTION

Our derivation for Eq. (8) largely follow the work of Kingma & Welling (2014); here for completeness we outline the key steps. First, Eq. (4) in Section 3.2 can equivalently be expressed as:

$$
\begin{aligned}
\mathcal{L}^{(i)} &= -D_{\mathrm{KL}}\Big( q_\phi(\mathbf{z}|\mathbf{a}^{(i)}, \mathbf{b}^{(i)}) \,\Big\|\, p_\theta(\,\mathbf{z}\,) \Big) + \mathbb{E}_{q_\phi(\mathbf{z}|\mathbf{a}^{(i)},\mathbf{b}^{(i)})}\Big[ \log p_\theta(\,\mathbf{b}^{(i)}|\mathbf{a}^{(i)}, \mathbf{z}\,) \Big] \\
&= -D_{\mathrm{KL}}\Big( q_\phi(\mathbf{z}|\mathbf{a}^{(i)}, \mathbf{b}^{(i)}) \,\Big\|\, p_\theta(\,\mathbf{z}\,) \Big) + \mathbb{E}_{p(\boldsymbol{\epsilon})}\Big[ \log p_\theta(\,\mathbf{b}^{(i)} \,|\, \mathbf{a}^{(i)}, \mathbf{z}\,) \Big],
\end{aligned} \tag{9}
$$

where $\mathbf{z} = g(\boldsymbol{\epsilon}, \mathbf{a}^{(i)}, \mathbf{b}^{(i)}, \phi)$, $\boldsymbol{\epsilon} \sim p(\,\boldsymbol{\epsilon}\,)$, and $g$ is some differentiable and invertible transformation. We can approximate $\mathcal{L}^{(i)}$ in Eq. (9) with a Monte Carlo estimator $\widetilde{\mathcal{L}}^{(i)}$:

$$
\widetilde{\mathcal{L}}^{(i)} = -D_{\mathrm{KL}}\Big( q_\phi(\mathbf{z}|\mathbf{a}^{(i)}, \mathbf{b}^{(i)}) \,\Big\|\, p_\theta(\,\mathbf{z}\,) \Big) + \frac{1}{L} \sum_{l=1}^{L} \log p_\theta(\,\mathbf{b}^{(i)} \,|\, \mathbf{a}^{(i)}, \mathbf{z}^{(i,l)}\,), \tag{10}
$$

where $\mathbf{z}^{(i,l)} = g(\boldsymbol{\epsilon}^{(i,l)}, \mathbf{a}^{(i)}, \mathbf{b}^{(i)}, \phi)$ and $\boldsymbol{\epsilon}^{(i,l)} \sim p(\,\boldsymbol{\epsilon}\,)$. Based on the RPDA functions $D = \{\, d(\mathbf{a}, \mathbf{b}; r) \mid r \in R \,\}$ and its relation preserving assumption that $q_\phi(\,\mathbf{z}|\mathbf{a}^{(i)}, \mathbf{b}^{(i)}\,) = q_\phi(\,\mathbf{z}|\mathbf{a}'^{(i)}, \mathbf{b}'^{(i)}\,)$ for $(\mathbf{a}'^{(i)}, \mathbf{b}'^{(i)}) = d(\mathbf{a}^{(i)}, \mathbf{b}^{(i)}; r), \forall r$ (see Section 3.3), we can express $\widetilde{\mathcal{L}}^{(i)}$ in Eq. (10) equivalently as $\widetilde{\mathcal{L}}_{\mathrm{RPDA}}^{(i)}$:

$$
\widetilde{\mathcal{L}}_{\mathrm{RPDA}}^{(i)} = -D_{\mathrm{KL}}\Big( q_\phi(\,\mathbf{z}|\mathbf{a}'^{(i)}, \mathbf{b}'^{(i)}\,) \,\Big\|\, p_\theta(\,\mathbf{z}\,) \Big) + \frac{1}{L} \sum_{l=1}^{L} \log p_\theta(\,\mathbf{b}^{(i)} \,|\, \mathbf{a}^{(i)}, \mathbf{z}^{(i,l)}\,), \tag{11}
$$

$$
\text{where} \qquad \mathbf{z}^{(i,l)} = g(\boldsymbol{\epsilon}^{(i,l)}, \mathbf{a}'^{(i)}, \mathbf{b}'^{(i)}, \phi), \quad \boldsymbol{\epsilon}^{(i,l)} \sim p(\,\boldsymbol{\epsilon}\,),
$$

$$
(\mathbf{a}'^{(i)}, \mathbf{b}'^{(i)}) = d(\mathbf{a}^{(i)}, \mathbf{b}^{(i)}; r^{(i)}), \quad r^{(i)} \sim \mathcal{U}(R).
$$

The first term in Eq. (11) is the KL-divergence from $p_\theta(\,\mathbf{z}\,)$ to $q_\phi(\,\mathbf{z}|\mathbf{a}'^{(i)}, \mathbf{b}'^{(i)}\,)$, which can be computed analytically given that we assume $p_\theta(\,\mathbf{z}\,) = \mathcal{N}(\mathbf{0}, \mathbf{I})$ and $q_\phi(\mathbf{z}|\mathbf{a}'^{(i)}, \mathbf{b}'^{(i)}) = \mathcal{N}(\mathbf{z}; \boldsymbol{\mu}^{(i)}, (\boldsymbol{\sigma}^{(i)})^2 \mathbf{I})$:

$$
-D_{\mathrm{KL}}\Big( q_\phi(\,\mathbf{z}|\mathbf{a}'^{(i)}, \mathbf{b}'^{(i)}\,) \,\Big\|\, p_\theta(\,\mathbf{z}\,) \Big) = \frac{1}{2} \sum_{j=1}^{2} \Big( 1 + \log((\sigma_j^{(i)})^2) - (\mu_j^{(i)})^2 - (\sigma_j^{(i)})^2 \Big). \tag{12}
$$

The likelihood function $p_\theta(\,\mathbf{b}^{(i)} \,|\, \mathbf{a}^{(i)}, \mathbf{z}^{(i,l)}\,)$ is defined as a multivariate Bernoulli distribution whose probability parameters $\mathbf{p}^{(i,l)}$ are computed from the neural network $f_\theta^{\mathrm{dec}}\big( f_\theta^{\mathrm{enc}}(\mathbf{a}^{(i)}), \mathbf{z}^{(i,l)} \big)$ and we have:

$$
\log p_\theta(\,\mathbf{b}^{(i)} \,|\, \mathbf{a}^{(i)}, \mathbf{z}^{(i,l)}\,) = \sum_k b_k^{(i)} \log p_k^{(i,l)} + (1 - b_k^{(i)}) \log(1 - p_k^{(i,l)}). \tag{13}
$$

By substituting Eq. (12) and (13) back in (11) and recall that $\mathbf{z}^{(i,l)} = g(\boldsymbol{\epsilon}^{(i,l)}, \mathbf{a}'^{(i)}, \mathbf{b}'^{(i)}, \phi) = \boldsymbol{\mu}^{(i)} + \boldsymbol{\sigma}^{(i)} \odot \boldsymbol{\epsilon}^{(i,l)}$, $p(\,\boldsymbol{\epsilon}\,) = \mathcal{N}(\mathbf{0}, \mathbf{I})$, $D = \{\, (\mathrm{rot}(\mathbf{a}, r), \mathrm{rot}(\mathbf{b}, r)) \mid r \in [0, 360) \,\}$, $L = 1$, we arrive at Eq. (8).

# B ADDITIONAL EXPERIMENTAL RESULTS

## B.1 ADDITIONAL MNIST RELATIONAL LEARNING EXPERIMENTS

Here, we provide additional experimental results based on the paired MNIST dataset constructed in Section 5.

### B.1.1 MNIST EXAMPLE WITH COUPLED RELATIONAL PROPERTY

First, to further test the robustness of the proposed method, we considered a scenarios where the underlying relational property is coupled with the absolute property. An example dataset $\mathbf{X}^2$ was constructed with the rotational relationship between each of the datapoint completely determined by its digit label: $\mathbf{X}^2 = \{\, (\mathbf{a}^{(i)} = \mathbf{x}^{(j,k)}, \mathbf{b}^{(i)} = \mathbf{x}^{(j,l+1)}) \mid j \sim \mathcal{U}([1..T]), \; k \sim \mathcal{U}([1..5]), l = $

$(k + \lfloor y^{(j)}/2 \rfloor) \mod 5$ }. In this case, it is possible to infer the relational property solely based on the image representation of the digit (absolute property of $\mathbf{a}^{(i)}$), for example, $\mathbf{a}^{(i)} \in [\text{'0'}, \text{'1'}] \rightarrow 0°$ (read: if $\mathbf{a}^{(i)}$ is recognized as either digit 0 or 1, the rotational relationship between $(\mathbf{a}^{(i)}, \mathbf{b}^{(i)})$ is $0°$) , $\mathbf{a}^{(i)} \in [\text{'2'}, \text{'3'}] \rightarrow 72°$, $\mathbf{a}^{(i)} \in [\text{'4'}, \text{'5'}] \rightarrow 144°$, $\mathbf{a}^{(i)} \in [\text{'6'}, \text{'7'}] \rightarrow 216°$, $\mathbf{a}^{(i)} \in [\text{'8'}, \text{'9'}] \rightarrow 288°$. The question then arise: is VRL capable of learning a decoupled relational property even when it is coupled with the absolute property? To test this idea, we trained VRL (with the same setup as that described in appendix A) on $\mathbf{X}^2$ but validated it on a hold-out dataset with a decoupled relational property (much like how $\mathbf{X}$ was constructed). The resulting scatter plot is shown in Fig. 7, where we can see that the approximated posterior $q_\phi(\mathbf{z} \mid \mathbf{a}, \mathbf{b})$ accurately cluster(discriminate) data with the same(different) rotational relationship together(apart). This result shows that VRL was indeed capable of learning a decoupled relational property irrespective of the digit representation of $\mathbf{a}^{(i)}$. If this were not the case, we would expect to see a scatter plot with heavily overlapped relationship labels since the validation dataset was constructed with random rotational relationships. Examples

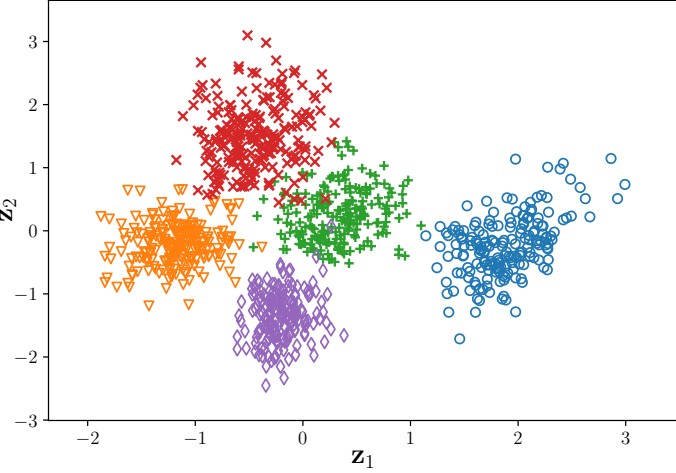

Figure 7: Scatter plot of the relational property (showing only the mean $\boldsymbol{\mu}$) of a hold-out dataset inferred by a VRL model that was trained on $\mathbf{X}^3$ (relationship labels: $\bigcirc : 0°$, $\triangledown : 72°$, $+ : 144°$, $\times : 216°$, $\Diamond : 288°$).

of images predicted by the learned likelihood function $p_\theta(\mathbf{b} \mid \mathbf{a}, \mathbf{z})$ are shown in Fig. 8, where Fig. 8a shows predicted images based on direct sampling in the latent space (shown in Fig. 7), and Fig. 8b shows examples of relational mappings. Figures 8a and 8b further corroborate our claim that

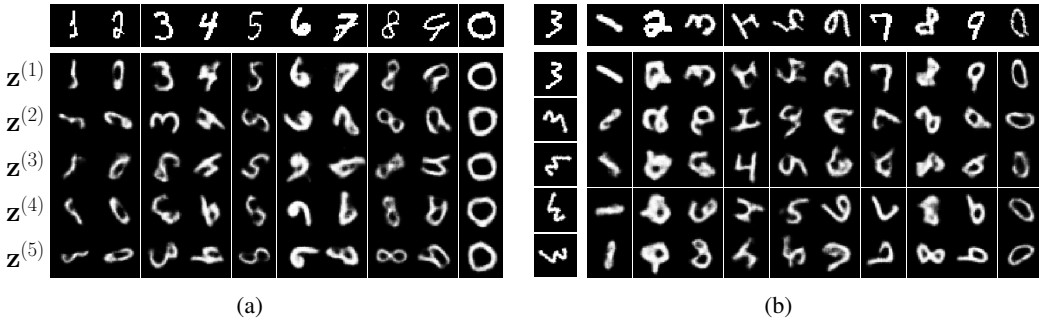

(a)        (b)

Figure 8: Examples of images predicted by a VRL model that was trained on $\mathbf{X}^2$: (a) images predicted from sampled latent variables (sampling the centroid of each cluster in Fig. 7: "$\bigcirc$" $\rightarrow \mathbf{z}^{(1)}$, "$\triangledown$" $\rightarrow \mathbf{z}^{(2)}$, "$+$" $\rightarrow \mathbf{z}^{(3)}$, "$\times$" $\rightarrow \mathbf{z}^{(4)}$, "$\Diamond$" $\rightarrow \mathbf{z}^{(5)}$); (b) examples of relational mappings of top row images by applying relationships inferred from pairs of source images $(\mathbf{a}_s, \mathbf{b}_s^{(r)})$ shown in the left-most column with $\mathbf{a}_s, \mathbf{b}_s^{(1)}, ..., \mathbf{b}_s^{(5)}$ arranged from top to bottom.

VRL is capable of learning a decoupled relational property and generalizing it to unseen data, e.g., VRL trained on $\mathbf{X}^2$ learned to rotate *any* digit by *any* amount despite not having seen most of the digit-rotation pairs during training. Note that this experiment is an example of *deterministic-mapping* discussed in Section 3.3, nevertheless VRL was able to utilize all paths in Fig. 2b effectively to learn about the data.

### B.1.2 MNIST EXAMPLE WITH MULTIPLE RELATIONAL PROPERTY

Next, we setup a more complex relational learning task that includes both rotational and resizing relationships between $(\mathbf{a}^{(i)}, \mathbf{b}^{(i)})$: each MNIST image $\mathbf{x}^{(i)}$ was augmented with five evenly-spaced rotations and three different resizing transformations to get $\{\, (\mathbf{x}^{(i,j,k)}, y^{(i)}) \mid i \in [1..T], j \in [1..5], k \in [1..3] \,\}$. Examples of augmented images are shown in Fig. 9, where top, middle, and bottom row images are resized by a factor of $\times 0.66$, $\times 1$, and $\times 1.5$, respectively. We considered a relational



Figure 9: Examples of a MNIST digit augmented with rotational and resizing transformations (from left to right: $\mathbf{x}^{(i,1,k)} : 0°$, $\mathbf{x}^{(i,2,k)} : 72°$, $\mathbf{x}^{(i,3,k)} : 144°$, $\mathbf{x}^{(i,4,k)} : 216°$, $\mathbf{x}^{(i,5,k)} : 288°$; from top to bottom: $\mathbf{x}^{(i,j,1)} : \times 0.66$, $\mathbf{x}^{(i,j,2)} : \times 1$, $\mathbf{x}^{(i,j,3)} : \times 1.5$).

learning task where each datapoint $(\mathbf{a}^{(i)}, \mathbf{b}^{(i)})$ has a decoupled rotational and/or resizing relational property; more specifically, we constructed a dataset $\mathbf{X}^3$ in the following way:

$$\{\, (\mathbf{a}^{(i)} = \mathbf{x}^{(j,k,m)}, \mathbf{b}^{(i)} = \mathbf{x}^{(j,l,n)}) \mid j \sim \mathcal{U}([1..T]), k, l \sim \mathcal{U}([1..5]), m \sim \mathcal{U}([1,2]), n \sim \mathcal{U}(V) \,\}$$

$$\text{where} \qquad V = \begin{cases} [1,2], & \text{if } m = 1 \\ [2,3], & \text{if } m = 2 \end{cases}$$

Note that in this case, $\mathbf{b}^{(i)}$ is either the same size as $\mathbf{a}^{(i)}$ or is $\times 1.5$ larger than $\mathbf{a}^{(i)}$, and there are a total of 10 different relationships between $(\mathbf{a}^{(i)}, \mathbf{b}^{(i)})$ in $\mathbf{X}^3$ (combinations of 5 rotational and 2 resizing transformations). We trained VRL on $\mathbf{X}^3$ with the same model setup as that described in appendix A and used the trained model to infer the relational property of a hold-out dataset. The inference result is shown in Fig. 10, where we can see that the approximated posterior $q_\phi(\,\mathbf{z} \mid \mathbf{a}, \mathbf{b}\,)$ accurately cluster(discriminate) data with the same(different) relationship together(apart). Examples of images predicted by the learned likelihood function $p_\theta(\,\mathbf{b} \mid \mathbf{a}, \mathbf{z}\,)$ are shown in Fig. 11, where Fig. 11a shows predicted images based on direct sampling in the latent space (shown in Fig. 10), and Fig. 11b shows examples of relational mappings. These results are consistent with the findings presented and discussed in Section 5.3.

### B.1.3 MNIST EXAMPLE WITH CONTINUOUS RELATIONAL PROPERTY

Finally, we present an example with a *continuous* relational property. Based on the MNIST dataset $\{\, (\mathbf{x}^{(i)}, y^{(i)}) \mid i \in [1..T] \,\}$, we constructed a paired dataset $\mathbf{X}^4$ in the following way:

$$\mathbf{X}^4 = \{\, (\mathbf{a}^{(i)} = \mathbf{x}^{(j)}, \mathbf{b}^{(i)} = \text{rot}(\mathbf{x}^{(j)}, r^{(i)})) \mid j \sim \mathcal{U}([1..T]), r^{(i)} \sim \mathcal{U}([0, 360)) \,\},$$

where $\text{rot}(\mathbf{x}^{(j)}, r^{(i)})$ rotates the image $\mathbf{x}^{(j)}$ by $r^{(i)} \in [0, 360)$ degrees. In this case, $\mathbf{b}^{(i)}$ is a random rotation of $\mathbf{a}^{(i)}$ and there is a continuous (and decoupled) rotational relationship between $(\mathbf{a}^{(i)}, \mathbf{b}^{(i)})$ in $\mathbf{X}^4$. We trained VRL on $\mathbf{X}^4$ with the same model setup as that described in appendix A and used the trained model to infer the relational property of a hold-out dataset. A scatter plot of the relational property infered by the approximated posterior $q_\phi(\,\mathbf{z} \mid \mathbf{a}, \mathbf{b}\,)$ is shown in Fig. 12, and examples of images predicted by the learned likelihood function $p_\theta(\,\mathbf{b} \mid \mathbf{a}, \mathbf{z}\,)$ are shown in Fig. 13, where Fig. 13a shows predicted images based on direct sampling in the latent space (denoted by markers "$\times$" in

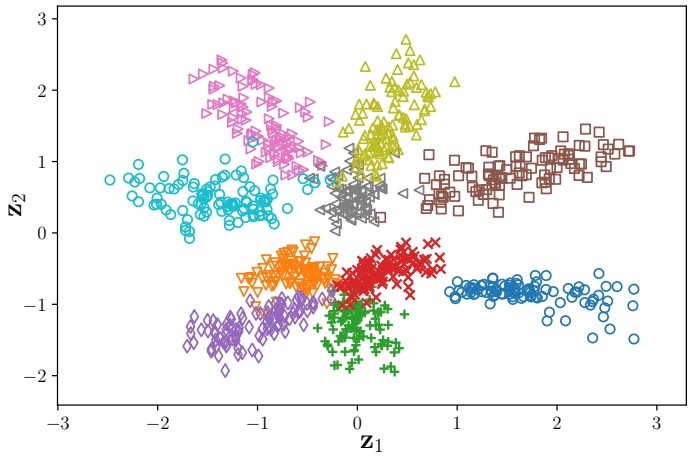

Figure 10: Scatter plot of the relational property (showing only the mean $\boldsymbol{\mu}$) of a hold-out dataset inferred by a VRL model that was trained on $\mathbf{X}^3$ (relationship labels: $\bigcirc$(blue)$:0°$, $\triangledown:72°$, $+:144°$, $\times:216°$, $\diamondsuit:288°$, $\square:0°$, $\times 1.5$, $\triangleright:72°$, $\times 1.5$, $\triangleleft:144°$, $\times 1.5$, $\triangle:216°$, $\times 1.5$, $\bigcirc$(cyan)$:288°$, $\times 1.5$).

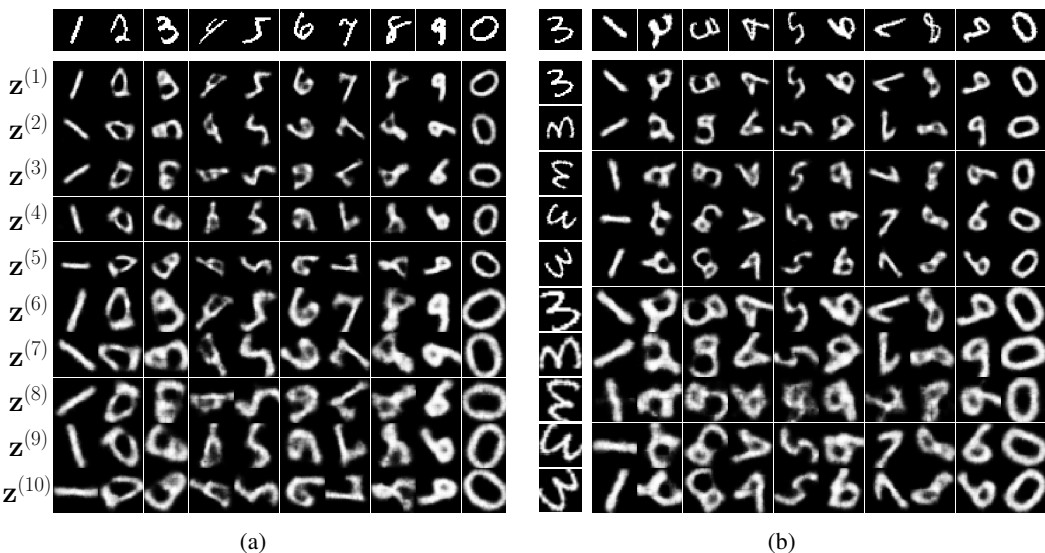

(a)             (b)

Figure 11: Examples of images predicted by a VRL model that was trained on $\mathbf{X}^3$: (a) images predicted from sampled latent variables (sampling the centroid of each cluster in Fig. 10: "$\bigcirc$"(blue)$\rightarrow \mathbf{z}^{(1)}$, "$\triangledown$"$\rightarrow\mathbf{z}^{(2)}$,"$+$"$\rightarrow\mathbf{z}^{(3)}$, "$\times$"$\rightarrow\mathbf{z}^{(4)}$, "$\diamondsuit$"$\rightarrow\mathbf{z}^{(5)}$, "$\square$"$\rightarrow\mathbf{z}^{(6)}$, "$\triangleright$"$\rightarrow\mathbf{z}^{(7)}$, "$\triangleleft$"$\rightarrow\mathbf{z}^{(8)}$, "$\triangle$"$\rightarrow \mathbf{z}^{(9)}$, "$\bigcirc$"(cyan)$\rightarrow\mathbf{z}^{(10)}$); (b) examples of relational mappings of top row images by applying relationships inferred from pairs of source images $(\mathbf{a}_s, \mathbf{b}_s^{(r)})$ shown in the left-most column with $\mathbf{a}_s, \mathbf{b}_s^{(1)}, ..., \mathbf{b}_s^{(10)}$ arranged from top to bottom.

Fig. 12), and Fig. 13b shows examples of relational mappings. From Fig. 12 and 13 we can see that VRL learned a decoupled relational property that encodes a continuous data (rotational) relationship; however, there is a small region in Fig. 12 with overlapping relational property that leads to an ambiguous interpretation of the rotational relationship ($120°$ vs. $240°$). This ambiguity is likely caused by compressing the continuous data (rotational) relationship down to a two-dimensional latent space, $\mathbf{z} \in \mathbb{R}^2$, and motivates us to adopt a higher-dimensional latent space, e.g., $\mathbf{z} \in \mathbb{R}^3$. Figure 14 shows inference result from repeating the previous experiment but with setting $\mathbf{z} \in \mathbb{R}^3$; we can see that VRL learned a three-dimensional relational property that unambiguously represents the underlying continuous data (rotational) relationship.

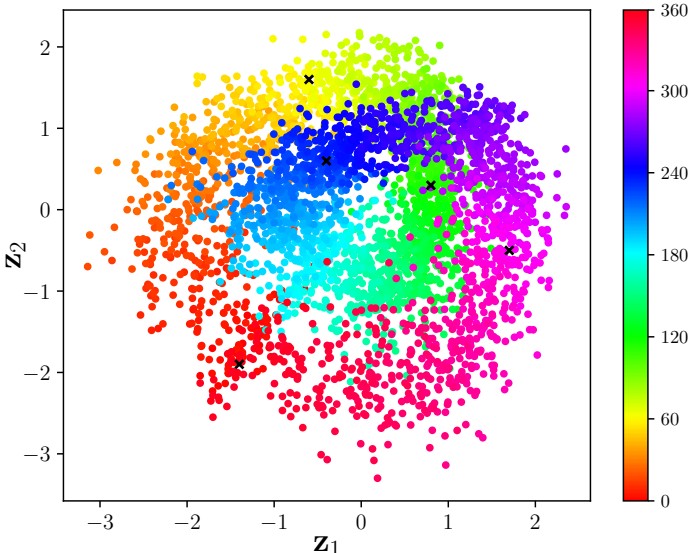

Figure 12: Scatter plot of the relational property (showing only the mean $\boldsymbol{\mu}$) of a hold-out dataset inferred by a VRL model that was trained on $\mathbf{X}^4$; each point is color-coded (best viewed in color) by the degrees of rotation between the corresponding datapoint (markers "$\times$" denote sampled latent varibles $\mathbf{z}^{(1)}, \ldots, \mathbf{z}^{(5)}$ used for image prediction in Fig. 13a).

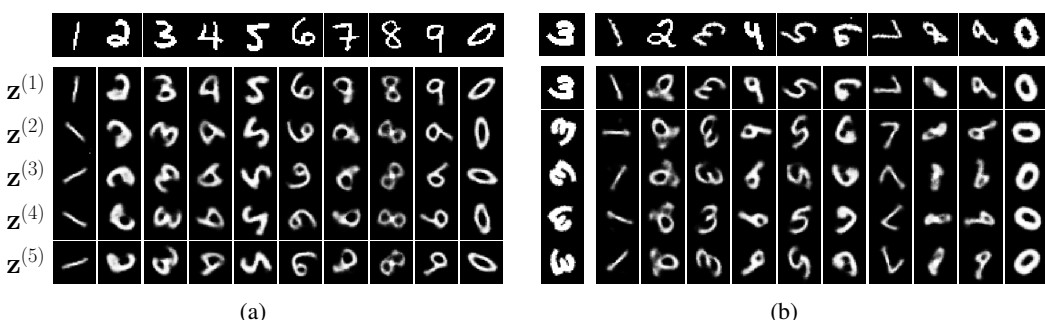

Figure 13: Examples of images predicted by a VRL model that was trained on $\mathbf{X}^4$: (a) images predicted from sampled latent variables (denoted by markers "$\times$" in Fig. 12); (b) examples of relational mappings of top row images by applying relationships inferred from pairs of source images $(\mathbf{a}_s, \mathbf{b}_s^{(r)})$ shown in the left-most column with $\mathbf{a}_s, \mathbf{b}_s^{(1)}, ..., \mathbf{b}_s^{(5)}$ arranged from top to bottom.

## B.2 YALE FACE RELATIONAL LEARNING EXPERIMENTS

### B.2.1 YALE FACE EXAMPLE WITH LEARNING EMOTION CHANGE

Here we present another relational learning example based on the Yale face dataset (Belhumeur et al., 1997). The Yale face dataset consists of 15 subjects, each with 8 facial expressions and 3 lighting configurations. We extracted three facial expressions (happy, surprised, sad) of each subject to form $\{\, \mathbf{x}^{(i,j)} \mid i \in [1..15], j \in [1..3] \,\}$. Examples of face images are shown in Fig. 15a, where we center-cropped, resized to $64 \times 64$, and normalized pixel values to be within $[0, 5]$. We constructed a dataset $\mathbf{X}^{F_e}$ where each datapoint $(\mathbf{a}^{(i)}, \mathbf{b}^{(i)})$ represents a subject with different facial expressions (*emotions*):

$$\mathbf{X}^{F_e} = \{\, (\mathbf{a}^{(i)} = \mathbf{x}^{(j,k)}, \mathbf{b}^{(i)} = \mathbf{x}^{(j,l)}) \mid j \sim \mathcal{U}([1..15]), \; k \sim \mathcal{U}([1..3]), \; l \sim \mathcal{U}([1..3] \setminus k) \,\}.$$

Our initial intention was to apply VRL to $\mathbf{X}^{F_e}$ to learn about the *"emotional change"* between $(\mathbf{a}^{(i)}, \mathbf{b}^{(i)})$ irrespective of the subject; however, because $\mathbf{X}^F$ is an extremely limited dataset that

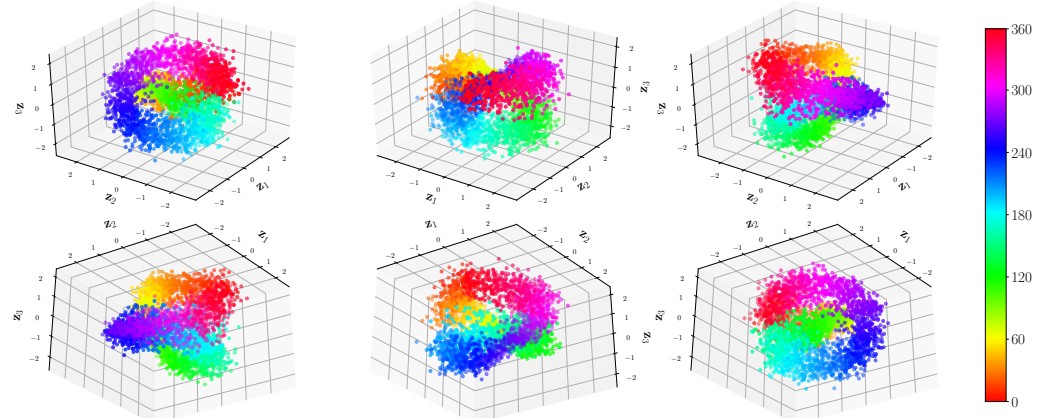

Figure 14: Scatter plot of the three-dimensional relational property (showing only the mean $\boldsymbol{\mu}$) of a hold-out dataset inferred by a VRL model (with $\mathbf{z} \in \mathbb{R}^3$) that was trained on $\mathbf{X}^4$; each plot shows a different vantage point of the 3D scatter plot, and each point is color-coded (best viewed in color) by the degrees of rotation between the corresponding datapoint.

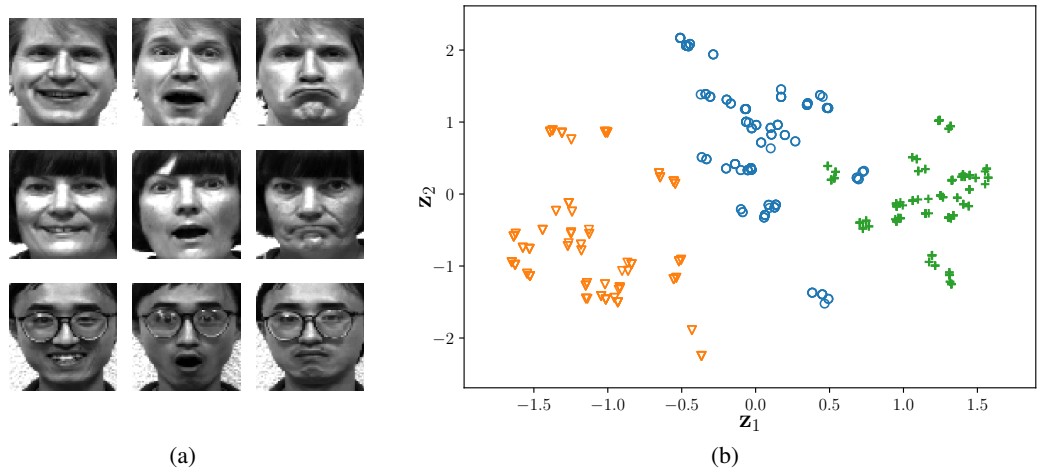

(a)                                                    (b)

Figure 15: Learning emotional changes among the Yale face dataset: (a) examples of subjects with different facial expressions: happy, surprised, and sad; (b) scatter plot of the relational property (showing only the mean $\boldsymbol{\mu}$) inferred by the approximated posterior (relationship labels: $\triangledown$: "happy-sad", $\bigcirc$: "happy-surprised", $+$: "surprised-sad").

consists of only 45 different images (15 subjects, each with 3 facial expressions), we settled for a less ambitious goal of learning an *undirected* emotional change, i.e., "happy→sad" = "sad→happy" and simply denoted as "happy-sad". In this case, there are three different emotional changes between $(\mathbf{a}^{(i)}, \mathbf{b}^{(i)})$ in $\mathbf{X}^F$: "happy-sad", "happy-surprised", "surprised-sad". To setup a VRL model for training, we adopted a two-dimensional latent variable $\mathbf{z} \in \mathbb{R}^2$ and let the prior $p_\theta(\mathbf{z})$ be the bivariate normal distribution. Since this is a real-valued dataset, we let $p_\theta(\mathbf{b}^{(i)} \mid \mathbf{a}^{(i)}, \mathbf{z}^{(i,l)})$ be a multivariate Gaussian distribution with a fixed diagonal covariance $\mathcal{N}(\mathbf{b}^{(i)}; \boldsymbol{\mu}_{\mathbf{b}}^{(i,l)}, \sigma^2 \mathbf{I})$ where $\boldsymbol{\sigma} = 0.1$ and $\boldsymbol{\mu}_{\mathbf{b}}^{(i,l)}$ is computed from a given $\mathbf{a}^{(i)}$ and $\mathbf{z}^{(i,l)}$ with an autoencoder-like neural network

$f_\theta^{\text{dec}}\left(f_\theta^{\text{enc}}(\mathbf{a}^{(i)}),\ \mathbf{z}^{(i,l)}\right)$:

$\quad f_\theta^{\text{enc}} : \mathbf{a}^{(i)} \to \text{Conv}(3\text{x}3\text{x}4) \to \text{Conv}(3\text{x}3\text{x}16) \to \text{Conv}(3\text{x}3\text{x}64) \to \text{FC}(50) \to \mathbf{h}^{(i)} \in \mathbb{R}^{50}$

$\quad f_\theta^{\text{dec}} : [\mathbf{h}^{(i)} \in \mathbb{R}^{50}, \mathbf{z}^{(i,l)} \in \mathbb{R}^2] \to \text{FC} \to \text{Conv}^T(3\text{x}3\text{x}64) \to \text{Conv}^T(3\text{x}3\text{x}16) \to \text{Conv}^T(3\text{x}3\text{x}4)$

$\qquad \to \text{Conv}(1\text{x}1\text{x}1) \to \boldsymbol{\mu}_{\mathbf{b}}^{(i,l)}$

and we have:

$$\log p_\theta(\,\mathbf{b}^{(i)} \mid \mathbf{a}^{(i)}, \mathbf{z}^{(i,l)}\,) = -\frac{1}{2}\frac{\left\|\mathbf{b}^{(i)} - \boldsymbol{\mu}_{\mathbf{b}}^{(i,1)}\right\|^2}{0.01} + \text{const.} \tag{14}$$

Again, we let the approximated posterior $q_\phi(\,\mathbf{z} \mid \mathbf{a}'^{(i)}, \mathbf{b}'^{(i)}\,)$ be a bivariate Gaussian distribution with a diagonal covariance $\mathcal{N}(\mathbf{z}; \boldsymbol{\mu}^{(i)}, (\boldsymbol{\sigma}^{(i)})^2\mathbf{I})$ where $\boldsymbol{\mu}^{(i)}$ and $\boldsymbol{\sigma}^{(i)}$ are the output of a neural network $f_\phi^q(\mathbf{a}'^{(i)}, \mathbf{b}'^{(i)})$:

$\quad f_\phi^q : [\mathbf{a}'^{(i)}, \mathbf{b}'^{(i)}] \to \text{Conv}(3\text{x}3\text{x}4) \to \text{Conv}(3\text{x}3\text{x}16) \to \text{Conv}(3\text{x}3\text{x}64) \to \text{FC}(4) \to [\boldsymbol{\mu}^{(i)}, \boldsymbol{\sigma}^{(i)}]$

Next, based on the premise that we are only interested in learning an undirected emotional change, we have $p_\theta(\,\mathbf{z} \mid \mathbf{a}, \mathbf{b}\,) = p_\theta(\,\mathbf{z} \mid \mathbf{b}, \mathbf{a}\,)$ and this prompted us to construct RPDA functions $D$ with random image rotation and swapping operations: $D = \{\,\text{swap}\,(\text{rot}(\mathbf{a}, r), \text{rot}(\mathbf{b}, r)) \mid r \in [0, 360)\,\}$ where

$$\text{swap}(\mathbf{a}, \mathbf{b}) = \begin{cases} (\mathbf{a}, \mathbf{b}), & p = 0.5 \\ (\mathbf{b}, \mathbf{a}), & p = 0.5 \end{cases}$$

Finally, we combine the above settings with $\mathbf{z}^{(i,l)} = \boldsymbol{\mu}^{(i)} + \boldsymbol{\sigma}^{(i)} \odot \boldsymbol{\epsilon}^{(i,l)}$, $p(\,\boldsymbol{\epsilon}\,) = \mathcal{N}(\mathbf{0}, \mathbf{I})$, $L = 1$, and substituting Eq. (12) and (14) back in (11) to derive the following lower bound estimator:

$$\widetilde{\mathcal{L}}_{\text{RPDA}}^{(i)} = \frac{0.01}{2} \sum_{j=1}^{2} \left(1 + \log((\sigma_j^{(i)})^2) - (\mu_j^{(i)})^2 - (\sigma_j^{(i)})^2\right) - \frac{1}{2}\left\|\mathbf{b}^{(i)} - \boldsymbol{\mu}_{\mathbf{b}}^{(i,1)}\right\|^2 + \text{const.} \tag{15}$$

$$\text{where} \quad (\boldsymbol{\mu}^{(i)}, \boldsymbol{\sigma}^{(i)}) = f_\phi^q(\mathbf{a}'^{(i)}, \mathbf{b}'^{(i)}), \quad \boldsymbol{\mu}_{\mathbf{b}}^{(i,1)} = f_\theta^{\text{dec}}\left(f_\theta^{\text{enc}}(\mathbf{a}^{(i)}),\ \mathbf{z}^{(i,1)}\right),$$

$$\mathbf{z}^{(i,1)} = \boldsymbol{\mu}^{(i)} + \boldsymbol{\sigma}^{(i)} \odot \boldsymbol{\epsilon}^{(i,1)}, \quad \boldsymbol{\epsilon}^{(i,1)} \sim \mathcal{N}(\mathbf{0}, \mathbf{I}),$$

$$(\mathbf{a}'^{(i)}, \mathbf{b}'^{(i)}) = \text{swap}\left(\text{rot}(\mathbf{a}, r^{(i)}), \text{rot}(\mathbf{b}, r^{(i)})\right), \quad r^{(i)} \sim \mathcal{U}([0, 360)).$$

The rest of the training setup remains the same as that described in appendix A. We trained VRL on $\mathbf{X}^{F_e}$ and then used the approximated posterior $q_\phi(\,\mathbf{z} \mid \mathbf{a}, \mathbf{b}\,)$ to infer the relational property of $\mathbf{X}^{F_e}$. The inference result is shown in Fig. 15b, where we can see that VRL learned a relational property that accurately differentiates emotional changes irrespective of the subject.

### B.2.2 YALE FACE EXAMPLE WITH LEARNING ILLUMINATION CONDITION CHANGE

Next, we present an experiment that learns relationship on "illumination condition changes" among the Extended Yale Face Database B (Georghiades et al., 2001). The Extended Yale Face Database B contains 16128 images of 28 human subjects under 9 poses and 64 illumination conditions. We extracted four illumination conditions (source of illumination: left, front, right, top) of each subject to form $\{\,\mathbf{x}^{(i,j)} \mid i \in [1..28], j \in [1..4]\,\}$. Examples of face images are shown in Fig. 16a, where we center-cropped, resized to $64 \times 64$, and normalized pixel values to be within $[0, 5]$. We constructed a dataset $\mathbf{X}^{F_l}$ where each datapoint $(\mathbf{a}^{(i)}, \mathbf{b}^{(i)})$ represents a subject with different illumination conditions (lightings):

$$\mathbf{X}^{F_l} = \{\,(\mathbf{a}^{(i)} = \mathbf{x}^{(j,k)}, \mathbf{b}^{(i)} = \mathbf{x}^{(j,l)}) \mid j \sim \mathcal{U}([1..28]),\ k \sim \mathcal{U}([1..4]),\ l \sim \mathcal{U}([1..4] \setminus k)\,\}.$$

Like the "learning undirected emotional changes" example presented in appendix B.2.1, we apply VRL to $\mathbf{X}^{F_l}$ to learn about the *"undirected illumination changes"* between $(\mathbf{a}^{(i)}, \mathbf{b}^{(i)})$ irrespective of the subject, i.e., "left→right" = "right→left" and simply denoted as "left-right". In this case, there are six different illumination changes between $(\mathbf{a}^{(i)}, \mathbf{b}^{(i)})$ in $\mathbf{X}^{F_l}$: "left-right"("L-R"), "front-top"("F-T"), "left-front"("L-F"), "left-top"("L-T"), "front-right"("F-R"), "right-top"("R-T"). We trained VRL on $\mathbf{X}^{F_l}$ with the *exact same* model and training setup as used in appendix B.2.1 and then used the

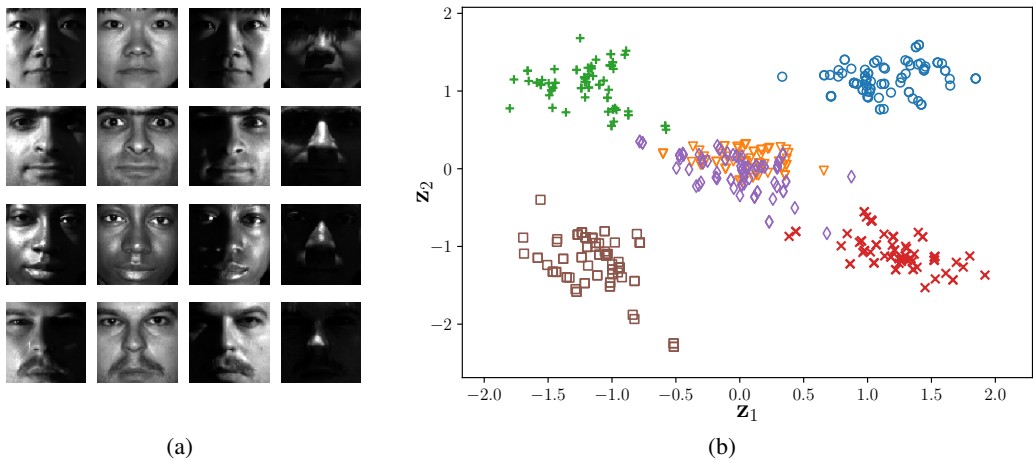

(a)                                          (b)

Figure 16: Learning illumination condition changes among the Yale face dataset: (a) examples of subjects with different illumination condition (source of illumination): left, front, right, and top; (b) scatter plot of the relational property (showing only the mean $\mu$) inferred by the approximated posterior (relationship labels: $\triangledown$: "left-right", $\diamondsuit$: "front-top", $\bigcirc$: "left-front", $+$: "left-top", $\times$: "front-right", $\square$: "right-top").

approximated posterior $q_\phi(\mathbf{z} \mid \mathbf{a}, \mathbf{b})$ to infer the relational property of $\mathbf{X}^{F_l}$. The inference result is shown in Fig. 16b where we can see that VRL correctly identifies four relationships ("L-T", "R-T", "F-R", "L-F" each represented by a distinct cluster in Fig. 16b) but collapses "F-T" and "L-R" into a single indistinguishable cluster. At first glance, this counterintuitive result—collapsing of "F-T" and "L-R"—seems to indicate that VRL was not able to learn meaningful representations for "F-T" and "L-R"; however, there is an elegant and logical explanation that justifies this unexpected result. Upon closer examination, we argue that it is indeed possible to combine the relationships "F-T", "L-R" without loss of information. In fact, we can give the combined relationship a meaningful name—"Opposite direction of illumination condition" or "Oppo." for short (in this interpretation it is more intuitive to rewrite "front" as "down"). With the newly formed compressed set of relationships $\mathbf{R}_c = \{\text{"L-T"}, \text{"R-T"}, \text{"F-R"}, \text{"L-F"}, \text{"Oppo."}\}$, it is easy to see that $\mathbf{R}_c$ is a valid set of relationships that fully and unambiguously characterizes the data (thus no loss of information) since: 1. any $(\mathbf{a}^{(i)}, \mathbf{b}^{(i)})$ can be characterized by *one and only one* relationship in $\mathbf{R}_c$; 2. for any given $\mathbf{a}^{(i)}$, each compatible relationship in $\mathbf{R}_c$ applied to $\mathbf{a}^{(i)}$ leads to a unique $\mathbf{b}^{(i)}$. And while we lost the identifiability of "F-T" and "L-R", combining them into "Oppo." is consistent with the relational learning goal—"Oppo." represents a *relative* relationship that does not depend on the illumination condition (absolute property) of each individual image. We stress that the proposed VRL method is not always guaranteed to learn a compact set of decoupled relationahips, and we will leave the investigation of this idea to a future work. Finally, we note that although the initial expection of learning a complete set of relationships is certainly reasonable and intuitive, it is quite surprising and unexpected that VRL is capable of discovering—in a completely unsupervised manner—a non-obvious set of relationships that is equally valid, and yet more compact.

To summarize the Yale face experiments presented in appendix B.2, we make the following two observations. First, when comparing our results with existing unsupervised leanring methods on face images (Song et al., 2007), we can see that existing methods cluster face images by its absolute property (e.g., subject identity, individual emotion, individual illumination condition, etc.) while the proposed VRL method clusters images by relationships (e.g., emotional change, illumination condition change). Second, it is worth noting that VRL does not learn the facial expression or illumination condition for each of $(\mathbf{a}^{(i)}, \mathbf{b}^{(i)})$ independently and then classify them based on their difference, but instead directly learns the emotional/lighting relationship between $(\mathbf{a}^{(i)}, \mathbf{b}^{(i)})$.

## C    ABLATION STUDY ON RPDA

In this work, we introduced relation-preserving data augmentation (RPDA) as a practical data augmentation strategy for overcoming the information-shortcut problem—an unique overfitting problem introduced by VRL (see Section 3.3). In order to evaluate the contribution of RPDA, we performed ablation study on the paired MNIST dataset $\mathbf{X}$ constructed in see Section 5.1. First, recall that in Section 5.2 we constructed RPDA functions $D$ with image rotation augmentations and optimized the following lower bound estimator (cf. Eq. (7)):

$$\widetilde{\mathcal{L}}_{\text{RPDA}}^{(i)} = \frac{1}{L} \sum_{l=1}^{L} \log p_\theta(\,\mathbf{b}^{(i)} \mid \mathbf{a}^{(i)}, \mathbf{z}^{(i,l)}\,) + \log p_\theta(\,\mathbf{z}^{(i,l)}\,) - \log q_\phi(\,\mathbf{z}^{(i,l)} \mid \mathbf{a'}^{(i)}, \mathbf{b'}^{(i)}\,), \quad (16)$$

$$\text{where} \qquad \mathbf{z}^{(i,l)} = g(\boldsymbol{\epsilon}^{(i,l)}, \mathbf{a'}^{(i)}, \mathbf{b'}^{(i)}, \phi), \quad \boldsymbol{\epsilon}^{(i,l)} \sim p(\,\boldsymbol{\epsilon}\,),$$

$$(\mathbf{a'}^{(i)}, \mathbf{b'}^{(i)}) = d(\mathbf{a}^{(i)}, \mathbf{b}^{(i)}; r^{(i)}), \quad r^{(i)} \sim \mathcal{U}(R).$$

In the first ablation study, we experimented with removing RPDA from the VRL training, which amounts to optimizing with the original lower bound estimator $\widetilde{\mathcal{L}}^{(i)}$ in Eq. (6). With the rest of the training setup remained unchanged (see appendix A), we trained VRL on $\mathbf{X}$ *without* RPDA and used the trained model to infer the relational property of a hold-out dataset; the inference result is shown in Fig. 17a. In the second ablation study, we repeated the VRL training, but instead of following the proposed RPDA strategy, we applied RPDA functions $D$ in a conventional data augmentation routine. More specifically, we optimized the lower bound $\widetilde{\mathcal{L}}^{(i)}$ in Eq. (6) over a minibatch of data augmented with $D$, $\mathbf{X'}^{M} = \{\,(\mathbf{a'}^{(i)}, \mathbf{b'}^{(i)}) \mid i \in [1..M]\,\}$, which leads to optimizing with the following lower bound estimator:

$$\widetilde{\mathcal{L}}_{\text{DA}}^{(i)} = \frac{1}{L} \sum_{l=1}^{L} \log p_\theta(\,\mathbf{b'}^{(i)} \mid \mathbf{a'}^{(i)}, \mathbf{z}^{(i,l)}\,) + \log p_\theta(\,\mathbf{z}^{(i,l)}\,) - \log q_\phi(\,\mathbf{z}^{(i,l)} \mid \mathbf{a'}^{(i)}, \mathbf{b'}^{(i)}\,), \quad (17)$$

$$\text{where} \qquad \mathbf{z}^{(i,l)} = g(\boldsymbol{\epsilon}^{(i,l)}, \mathbf{a'}^{(i)}, \mathbf{b'}^{(i)}, \phi), \quad \boldsymbol{\epsilon}^{(i,l)} \sim p(\,\boldsymbol{\epsilon}\,),$$

$$(\mathbf{a'}^{(i)}, \mathbf{b'}^{(i)}) = d(\mathbf{a}^{(i)}, \mathbf{b}^{(i)}; r^{(i)}), \quad r^{(i)} \sim \mathcal{U}(R).$$

With the rest of the training setup remained unchanged (see appendix A), we trained VRL on $\mathbf{X}$ with maximizing $\widetilde{\mathcal{L}}_{\text{DA}}^{(i)}$ and used the trained model to infer the relational property of a hold-out dataset; the inference result is shown in Fig. 17b. Comparing the results from both ablation studies, shown in

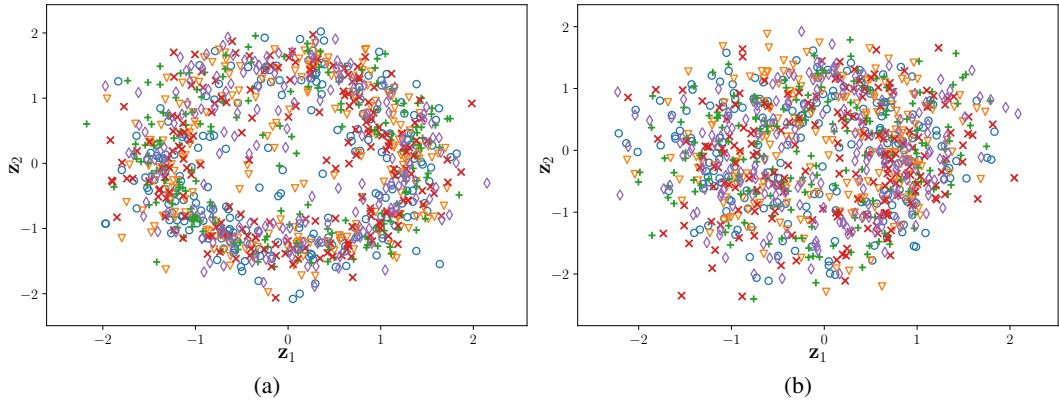

(a)                                                        (b)

Figure 17: Scatter plots of the relational properties (showing only the mean $\boldsymbol{\mu}$) generated from VRL ablation studies (relationship labels: $\bigcirc : 0°$, $\triangledown : 72°$, $+ : 144°$, $\times : 216°$, $\Diamond : 288°$): (a) relational property inferred by a VRL model that was trained without RPDA; (b) relational property inferred by a VRL model that was trained with applying RPDA functions $D$ in a conventional data augmentation routine.

Fig. 17, with inference result from VRL training *with* RPDA, shown in Fig. 4a, we can make the

following observations: first, RPDA is a critical component necessary for VRL to learn a meaningful and decoupled relational property, especially when flexible function approximations such as deep neural networks are used; second, by looking at the difference between $\widetilde{\mathcal{L}}_{\text{DA}}^{(i)}$ and $\widetilde{\mathcal{L}}_{\text{RPDA}}^{(i)}$ (observe that $\widetilde{\mathcal{L}}_{\text{DA}}^{(i)}$ applies RPDA functions $D$ to every term in Eq. (17), whereas $\widetilde{\mathcal{L}}_{\text{RPDA}}^{(i)}$ applies $D$ only to selected terms in Eq. (16) according to the RPDA strategy), we can draw the conclusion that the improvements brought by RPDA (and its key innovation) comes not from *what* data augmentation functions are applied but *how* they are applied.

## D  ADDITIONAL REMARKS

### D.1  REMARKS ON VRL-PGM

The central idea of the proposed VRL method is to encapsulate the relational learning problem with a probabilistic graphical model—VRL-PGM—and then formulate various relational processing tasks as performing inference and learning in the graphical model. Here we discuss aspects of the original relational learning problem (see Section 2) that differ from the proposed VRL-PGM (see Section 3.1). First, the original problem specifies that the relational property be decoupled from *both* $\mathbf{a}$'s and $\mathbf{b}$'s absolute properties; however, the latent variable $\mathbf{z}$ that is used to represent relational property in VRL-PGM is only independent of $\mathbf{a}$ but not $\mathbf{b}$. Second, we note that the original problem is inherently undirected with no cause-effect relationship between $\mathbf{a}$ and $\mathbf{b}$, whereas VRL-PGM is based a directed acyclic graph (DAG) that artificially introduces conditional dependency between $\mathbf{a}$ and $\mathbf{b}$. However, we argue that the application of VRL does not require the true conditional dependency between $(\mathbf{a}, \mathbf{b})$ be known in advance only that it is maintained consistently throughout learning and inference, i.e., VRL can be applied in the same way to learn about the relational property between $(\mathbf{b}, \mathbf{a})$, where we swap $\mathbf{a}$ and $\mathbf{b}$. The above-mentioned two discrepencies represent the compromises we made with adopting VRL-PGM in exchage for a riorous and tractable method for learning a decoupled relational property. We can futher view VRL's optimization challenges—deterministic-mapping and information-shortcut (introduced in Section 3.3)—as the consequence of these compromises: *deterministic-mapping* can viewd as caused by the causal relationship VRL-PGM introduced between $\mathbf{a}$ and $\mathbf{b}$, and *information-shortcut* can be viewed as caused by the causal relationship VRL-PGM introduced between $\mathbf{z}$ and $\mathbf{b}$.

### D.2  REMARKS ON INDEPENDENCE BETWEEN $\mathbf{z}$ AND $\mathbf{a}$

Recall that the assumption of independence between $\mathbf{z}$ and $\mathbf{a}$ is central to the VRL learning of a decoupled relational property, and in this work we rely on the construction of VRL-PGM to support such assumption; however, the learning objective (Eq. 4) derived from VRL-PGM does not explicitly force $\mathbf{z}$ to be independent of $\mathbf{a}$ (nor penalize learning a dependent $\mathbf{z}$). Here, we explain how optimizing Eq. 4 affects the learning of independent $\mathbf{z}$ and $\mathbf{a}$. The main learning objective of VRL is described in Eq. 4, where we can further dissect its various terms and observe that the learning is heavily guided by maximizing the likelihood term $\log p_\theta(\mathbf{b}^{(i)}|\mathbf{a}^{(i)}, \mathbf{z})$ since this is the only term that is constrained by the data while all other terms in Eq. 4 can be viewed as regularization on the unobserved $\mathbf{z}$. Next, when maximizing $\log p_\theta(\mathbf{b}^{(i)}|\mathbf{a}^{(i)}, \mathbf{z})$ we are only learning to predict $\mathbf{b}^{(i)}$ from $\mathbf{a}^{(i)}$ and $\mathbf{z}^{(i)}$; and since $\mathbf{a}^{(i)}$ is already been conditioned on, there is no incentive for $\mathbf{z}^{(i)}$ to learn redundant (dependent) information from $\mathbf{a}^{(i)}$. This effect is further "encouraged" when the learning objective Eq. 4 also includes additional regularization terms on $\mathbf{z}$ for learning a compact representation that "squeeze-out" any redundant information that does not help with predicting $\mathbf{b}^{(i)}$. In the information flow diagram shown in Fig. 2a one may argue that since $\mathbf{a}^{(i)}$ also propagate information through the latent variable $\mathbf{z}$ it may introduce dependency between $\mathbf{a}$ and $\mathbf{z}$; however, the information propagated from $\mathbf{a}^{(i)}$ through $\mathbf{z}^{(i)}$ is mainly used to maximize the likelihood term for predicting $\mathbf{b}^{(i)}$, and since there is already a direct propagation path from $\mathbf{a}^{(i)}$ to $\mathbf{b}^{(i)}$ (as part of the likelihood function $p_\theta(\mathbf{b}^{(i)}|\mathbf{a}^{(i)}, \mathbf{z})$) there is, again through regularization on $\mathbf{z}$, no incentive for $\mathbf{z}$ to carry redundant information from $\mathbf{a}$ (only decoupled relationship information derived from $\mathbf{a}$ and $\mathbf{b}$). In short, while there are no explicit penalties on learning dependent $\mathbf{z}$ and $\mathbf{a}$ in VRL's learning objective, the independence is naturally encouraged through the interplay between the different terms in Eq. 4. This effect is also corroborated by our experimental results where we can see that VRL can indeed learn a decoupled (independent) relational property $\mathbf{z}$ through optimizing Eq. 4.

Ideally, the learning of independent $\mathbf{z}$ and $\mathbf{a}$ can be achieved through VRL's learning mechanism discussed above; however, in practice there may be numerous reasons that could cause this to fail, such as insufficient training data, failure to reach the global optimum, non-identifiability of the model, etc. Therefore, for pragmatic reasons, it may be desirable to explicitly safeguard against introducing dependency between $\mathbf{z}$ and $\mathbf{a}$. Here, we propose a straightforward extension to VRL for achieving this goal: we can append any non-positive function that measures the dependency between $\mathbf{a}$ and $\mathbf{z}$ with maximum attained when they are independent to Eq. (4) without invalidating the lower bound. For example, we can append the negative mutual information between $\mathbf{z}$ and $\mathbf{a}$, $-I(\mathbf{z}, \mathbf{a}) = -D_{\mathrm{KL}}(p_\theta(\mathbf{z}, \mathbf{a}) \parallel p_\theta(\mathbf{z})p_\theta(\mathbf{a}))$, to Eq. (4) to obtain:

$$\mathcal{L}^{(i)} = \mathbb{E}_{q_\phi(\mathbf{z}|\mathbf{a}^{(i)}, \mathbf{b}^{(i)})} \Big[\log p_\theta(\mathbf{b}^{(i)}|\mathbf{a}^{(i)}, \mathbf{z}) + \log p_\theta(\mathbf{z}) - \log q_\phi(\mathbf{z}|\mathbf{a}^{(i)}, \mathbf{b}^{(i)})\Big] - I(\mathbf{z}, \mathbf{a}). \quad (18)$$

And since $I(\mathbf{z}, \mathbf{a}) \geq 0$ and $I(\mathbf{z}, \mathbf{a}) = 0$ *if and only if* $\mathbf{z}$ and $\mathbf{a}$ are independent, the addition of $-I(\mathbf{z}, \mathbf{a})$ in Eq. (18) not only does not invalidate the original bound in Eq. (4), but it also does not sacrifice the quality of the lower bound ($\mathbf{z}$ and $\mathbf{a}$ are independent in VRL-PGM). We will leave the investigation of this idea to a future work.

### D.3 REMARKS ON RPDA

Here we would like to comment on the practical applicability of the proposed RPDA stratey. More specifically, we would like to convey the idea that in many practical problem settings, the RPDA functions $D$ can be designed without any knowledge of the underlying relational property. For example, as we have explained in Sec. 3.3, in many computer vision applications, rotation invariant is a desirable property for the learned model; for example, in spectral imaging applications, oftentimes the orientation of the images are not preserved or not enforced (only that they are consistent between the same paired images) (Ronneberger et al., 2015). In such problem setting, we can safely use image rotation function for constructing $D$. Another example may be: for a discrete time-series data $\mathbf{a}[t], \mathbf{b}[t]$ that represent the input and output of a linear time-invariant (LTI) system (commonly assumed in signal processing and control theory (Oppenheim & Schafer, 2009)), and we want to learn a relational property that characterize the system's impulse response. We have $\alpha\mathbf{b}[t - \tau] = \alpha\mathbf{a}[t - \tau]$, $\forall \alpha \in \mathbb{R}, \tau \in \mathbb{Z}$, and we can construct $D$ with $d(\mathbf{a}[t], \mathbf{b}[t]; \alpha, \tau) = (\alpha\mathbf{a}[t - \tau], \alpha\mathbf{b}[t - \tau])$, $\alpha, \tau \in R = \mathbb{R} \times \mathbb{Z}$. In all of the above examples, the construction of RPDA functions $D$ reflects our prior knowledge and belief of the underlying system and not based on the data relationships; therefore, in many instances, RPDA can be designed without any knowledge of the underlying relational property. However, we would also note that RPDA is not central to the theory of the proposed method—VRL can be applied without RPDA—but rather a practical data augmentation strategy for addressing a unique optimization challenge (information-shortcut) of VRL learning. In all of our experiments, we find RPDA to be effective and crucial in overcoming the information-shortcut problem (as illustrated in appendix C). But just like any data augmentation, this is problem dependent and we advocate to start without RPDA and only apply it when suspecting information-shortcut occurs.

### D.4 REMARKS ON EXPERIMENTAL RESULTS

Here we would like to give an overall summary and make general remarks on the experimental results we have presented in this work (included in Sec. 5 and appendix B). First, we proposed four MNIST experiments that represent a diverse set of relational learning problems: decoupled relational property (Sec. 5.1), coupled relational property (appendix B.1.1), multiple relational properties (appendix B.1.2), and continuous relational property (appendix B.1.3). Although these experiments are introduced with well-controlled ground-truth relationships (so that we can easily validate and interpret the results), the application of the proposed VRL method is completely unsupervised without *any* prior knowledge of the underlying relationships. Furthermore, we deliberately design and successfully solve all four MNIST tasks using the *exact same* model and training setup despite each experiment represents a very different relational learning scenario (discrete vs. continuous, coupled vs. decoupled). In addition, we presented two Yale face tasks with high-level perception relationships: change of emotions (appendix B.2.1), and change of illumination conditions (appendix B.2.2). Again, the application of VRL to the two Yale face tasks is completely unsupervised and we deliberately design and successfully solve these two tasks using the *exact same* model and training setup.

When comparing our model and training setups between the MNIST and Yale face experiments, their differences can all be attributed to the need for accommodating different data types, e.g., increasing nework size for larger face images (vs. smaller MNIST images), modifying learning objective to adapt real-valued Yale dataset (vs. binay valued MNIST dataset), and not for accomodating different relationships. Taking this into consideration, our results shows that we have solved both class of problems with the *exact same* principled method despite each class of problems represents a very different kind of relationships (the relationships in MNIST are geometric whereas the relationahips in Yale are high-level perception, e.g., sentiment, external environmental factors). We believe our results further demonstrates that the proposed VRL method is robust, stable, and generalizable to many different relationships.

