# OpenReview forum: "Relational Learning with Variational Bayes"
_ICLR.cc/2021/Conference — Reject_

### Official Review · AnonReviewer2 · 2020-10-27
**Interesting problem in need of more theory and experiments**

**Rating:** 6
**Confidence:** 3

**Review:**

Relational learning is an important capability exhibited by humans of learning relations between objects.  This work considers a fairly general setup for relational learning and addresses learning of the relation through a method based on variational inference.

I am not familiar with the VAE literature; having said that, the method seems novel.  The problem setup is so general that it is unlikely to be novel, but the main technical contribution of the paper appears to be the derivation of the objective (4).  Of course, given that the objective is a lower bound of the probability we want to optimize, and since no theory is provided on the quality of the approximation, it is important to have strong demonstrations of the method in which it is shown to have unique advantages compared to existing approaches.  Unfortunately, the paper falls short in this regard, as I will go into more detail.

The equation (4) does seem plausible and it is rather elegant.  Note that (4) can also be seen as likelihood + regularization on z + entropy of z, hence it encourages latent variables z that are not too large according to the regularization, while the entropy term prevents z from collapsing to the mode.  This makes sense, but at the same time it is difficult to see how independence between z and a would be enforced using this objective function.

The other technical contribution is the identification of optimization issues via the "information shortcut" and "deterministic mapping" issues.  This is very insightful.  The idea of RPDA is interesting, and in many cases it could be applied at least via data-augmentation.

The main demonstration is based on learning rotation angles in MNIST.  The relationship of "A is 30 degrees rotated compared to B" does qualify as a relation, albeit one that seems fairly artificial.  (EDIT: since discussion, I retract the following in brackets).  [More worrisome is that the example used, if I understand correctly, does not follow the causal model adopted by the authors.   The model assumes that a and z cause b.  However, the example has b as the MNIST label.  This is causally incorrect because for handwritten digits, the label is the cause of the image, not vice-versa.  Presumably the causal assumptions should be correct for the model to work well?  It would be helpful if the authors could comment on this.]

The Yale face dataset in the supplement seems a better fit to the causal assumptions.

However neither example is satisfactory in showing a unique benefit of this approach that cannot be already obtained by domain-specific methods, such as spatial transformer networks in the case of learning image rotations/scaling (https://papers.nips.cc/paper/5854-spatial-transformer-networks.pdf) and emotion learning ML methods for facial images (https://www.sciencedirect.com/science/article/pii/S1877050917327679).  Perhaps the authors could propose an application that would showcase the advantages of the approach, even if it is currently infeasible to obtain results?

If we contrast the relational learning approach to the existing body of work on learning disentangled representations, the main difference is that in existing representation learning work, one learns the latent variable _z_ from _a_ only.  But here, _z_ can be learned from both _a_ and _b_ .  This is obviously an important difference, but one could reduce the problem of learning _z_ from _a_, _b_ to methods that learn _z1_ from _a_ and _z2_ from _b_ separately.  For example, in learning the rotation between _a_ and _b_ (both MNIST digits), one could reduce it to learning the orientation of _a_, learning the orientation of _b_ separately, and then subtracting the orientations.  Of course, this seems to be a property of the examples used rather than a general fact.  If one considers more sophisticated relational learning problems such as properties and objects, then it may no longer be reducible.  Yet this again indicates that the examples used in the paper fail to show the uniqueness of the proposed approach.

### Pros:
 * interesting problem with applications to psychology
 * insightful analysis of the optimization involved and remedies for two pitfalls

### Cons:
 * toy examples inadequate to show novelty and generalizability of the method
 * not obvious to me how the objective function encourages independence of _z_ and _a_.
 * do we have any confidence that this method would work in a more complicated problem?

### Recommendation:
The paper starts an interesting line of development but still seems to need more work in either the theory, the sophistication of demonstrations, or both, before one could make a convincing case that it adds something beyond similar approaches in the literature.  Hence I recommend rejection of this paper.  The authors are encouraged to look into adding more sophisticated and unique examples, and further analyzing their method.

### Modifications since discussion
 * retracted objection about inappropriateness of MNIST example
 * raised score from 5 to 6

---

> ### Author Response · Authors · 2020-11-12
> **Author response to Reviewer 2 review**
>
> Dear Reviewer:
>
> We would first like to express our deep appreciation for your time and insightful comments. Please find our response to your concerns (referring as R2) in the order they are raised:
>
> 1.	We viewed our main contribution as the proposal of a general and flexible variational learning framework for relational learning: including the proposal of VRL-PGM; a learning objective (variational lower bound) derived from first principle; and, a practical, end-to-end, completely unsupervised learning method. The proposed VRL-PGM that artificially introduces causal relationship by using a directed acyclic graph reflects our priority and compromise for representing the original abstract relational learning problem with a PGM: we sacrificed some identifiability of the original abstract problem but we gained a rigorous and mathematical tractable statistical model (reasoning and learning in a directed graph is much easier compared to undirected graph) while achieving our primary objective of learning an independent (decoupled) relational property. We have added a discussion of the proposed VRL-PGM and its connection to the original relational learning problem in the appendix. Please also see our response #1 to R1.
> 2.	The independence between latent variable z and a is an inherent property of the proposed VRL-PGM and not something we need to enforce explicitly. Please allow us to give an intuitive explanation on this point in terms of decomposing the objective (as noted by R2) into “likelihood + regularization on z + entropy of z”. We can focus the discussion on the likelihood term p(b|z,a) since this is the only term that is constrained by the data a and b (other terms can be viewed as regularization on the unobserved z). In the maximization of the likehood p(b|z,a), the objective is to learn a mapping to fit ONLY data b given a and z; therefore, there is no incentive for z to learn redundant (dependent) information from a; this effect is further “enforced” when the learning objective includes additional regularization term on z to learn a compact representation. In the information flow diagram shown in Fig. 2 of our submission one may argue that since a also propagate information through the latent variable z it may introduce dependency between a and z (in the posterior p(z|a,b), z and a are no longer independent!); however, the information propagated from a through z is mainly used to maximize the likelihood function for predicting b, and since a already provide a direct propagation path to b in p(b|a,z) there is, again (via regularization on z), no incentive for z to carry redundant absolute property information from a (only need to carry the decoupled relationship information from a and b). In summary, the independence between z and a is an inherent property of VRL-PGM, the variational lower-bound (learning objective) we derived based on VRL-PGM from first principle naturally disincentives learning a z that carries redundant (dependent) information of a without the need to explicitly enforce it. However, we understand R1’s concerns for taking a more conservative viewpoint and wanting to explicitly safeguard against learning dependency between z and a. To this end, we would like to present another advantage of the proposed VRL-PGM and the derived variational lower bound: we can add any  non-positive function that measures the dependency between a and z with maximum attained when they are independent to Eq. 4 without invalidating the lower bound. For example, we can safely add the mutual information between z and a, -I(z, a), to Eq. (4) where since I(z,a)>=0 and I(z,a)=0 iff z and a are independent (which they are in VRL-PGM!). Please let us know if you would like us add this discussion to the submission. Lastly, we want to emphasize that the goal of relational learning is to learn a relational property (represented by latent variable z) that is independent from the absolute property of a. Relational property z will necessarily derive relationship information from both a and b but should not depend on the absolute property of a (any information that describe a alone).
>
> (To be continued in the appending post)

---

> > ### Author Response · Authors · 2020-11-12
> > **Author response to Reviewer 2 review (Cont.)**
> >
> > 3.	We do not fully understand R2’ comment that “example used does not follow the causal model framework” and we look forward to further clarification on the comment and and provide you with a detailed explanation. R2 correctly stated that in the proposed VRL-PGM model a and z causes b. We note that the goal of variational inference is to learn an approximate posterior p(z|a,b) that infers z from a, b. The label of a and b are only used to generate the training dataset and the actual learning is completely unsupervised and do not have access to the labels. The annotation (relationship labels) we showed in the experiment results are only added to help with interpretation and visualization. The actual (2D) output of the approximated posterior are only the location of the points and not the labels.
> >
> > 4. We interpret the remaining concerns raised by R2 as around the question of whether we can solve the presented MNIST relational learning example with existing methods. More specifically, R2 raised the possibility of first learning a reduced representation (possibly with domain specific method that extract relevant information) of a and b, and subsequently reason within the reduced representation. We have spent significant effort pursuing this idea in our early investigation of the problem, please allow us to share our experience along this line of thinking in the form of the following thought experiment: Let us focus the discussion on the MNIST example presented in the main text: a and b represent the same MNSIT digit but each with random rotation selected from five evenly spaced rotations between 0 and 360 degrees (this is clearly specified in section 5.1). In this case, it is clear that there are only 5 uniquely defined rotation relationship between a and b (b is related to a by rotating 0, +72 (-288), +144 (-216), +216 (-144), +288 (-72)). Let us assume that we know in advance that the underlying relationship is rotational (which we do not assume in our work) and use domain specific method, such as the ones suggested by R2, to extract the individual rotation information for a and b. Let us further assume we can achieve this goal perfectly and map each of a and b to a SINGLE point in the latent space (out of five point where each point in the latent space represent different rotations). Now we face the question of how to reason within the latent space that learns 5 unique points corresponding to distinct rotations. A common practice for reasoning within the latent space is to perform vector arithmetic (think of word2vec in NLP and R2’s suggestion for “subtracting the orientations”). Let us investigate this idea further: after performing the vector arithmetic operation by representing (a,b) as a vector b-a (or a-b) in the learned latent space, we will have 25 unique vectors (defined by starting and end point and include vectors with zero length) that represent all possible combinations of (a,b) (Note that a and b are both random rotations of the same digit and  each of them will equally likely be mapped to any of the 5 rotation point in the latent space; thus, we have 5*5=25 different combination between a and b). Finally, we are left with the question on how to infer the rotation relationship (we know there are only 5) from the 25 unique vectors. One may be tempted to categorized these 25 vectors into 5 category based on their features (such as length, direction, etc) but will soon realize that this is not at all a trivial task (this can be intuitively understood by experimenting with placing five point on the paper, construct 25 vectors by making all possible connection with direction, and try to come up with a consistent rule to categorize the 25 different vectors into five category). In fact, any such attempt will inevitably introduce strict requirement on the placement (location) of the five rotation points and this location requirement is neither guaranteed nor easily implemented by any domain specific methods especially when the learning is unsupervised. We hope this discussion illustrates the provided seemly simple MNIST relational learning examples are in fact very challenging for existing unsupervised methodology and, therefore, we do not believe existing method is capable of solving this problem, let alone more complicated MNIST relational learning problems presented in the appendix with coupled relational property, multiple relational property, and continuous relational property. The same reasoning we provided above also carries over to answer the question of why exiting method does not work with emotion detection for face image, reasoning with reduced representation of a and b (noted by R2 as "z1 from a and z2 from b").
> >
> > (To be continued in the appending post)

---

> > > ### Author Response · Authors · 2020-11-12
> > > **Author response to Reviewer 2 review (Cont.)**
> > >
> > > 5. Of course, the discussion in our response #4 only represents one specific implementation/methodology but we can derive a deeper understanding of the fundamental challenge from the above example and make a more general observation on why relational learning problem (including the provided MNIST example) are difficult to solve with existing methods. To the best of our knowledge, most existing unsupervised method learns relationship by deriving relationship information from absolute properties (this includes graph-based method that uses edges to represent relationships). By definition, this derived relationship information necessarily couples with (depends on) the absolute property of the data; which also means that no matter how one may reduce the relational learning problem (including the MNIST examples) and try to solve it with exiting method, the fundamental relational learning problem of decoupling relational property from absolute property (whether the absolute property are from the raw data, reduced representation of the raw data, or the output of domain-specific method) will always be present and needs to be addressed.  Take the thought experiment presented in our response #4 for example, even after assuming we can successfully represent a pair of rotated MNIST digits by a vector in the latent space (an open question of whether it can be done in the first place) and thus significantly simplified the data representation, we are still left with the challenging question on how to reason/categorize the 25 unique vectors and deduce that there are only 5 underlying relationships. This challenge can be further understood in terms of absolute vs. relational property: these 25 vectors represent both absolute property and relational property of a and b (the starting point and end point of each vector are the absolute properties of a and b which represents individual rotation features of a and b whereas the direction and length of the vector representation encapsulate the relationship.) and it is not an easy task to decouple these two properties. A key insight into why our proposed method is suitable for solving the relational learning problem is recognizing that we treat and learn relationship as a stand-alone entity (latent variable z) and further design our VRL-PGM so that z is independent from a.
> > >
> > > In summary, we DO believe the provided MNIST relational learning examples represent novel and unique questions and clearly demonstrate the relational learning challenge that are not easily solvable with existing methods. In addition to the MNIST example presented in the main text, we include three additional MNIST relational learning example in the appendix each with increasing complexity (coupled relational property, multiple relational property, and continuous relational property). We demonstrate the robustness of the proposed method by successfully achieving satisfactory results on all four problems with the exact same model setup. To further demonstrate the generalization ability of the proposed method on a more complicated problem, we presented a face relational learning example and successfully solved it with a very similar setup as in our MNSIT example where we only made necessary adjustment to the model setup to accommodate the different types of data (e.g., larger image and continuous vs binary data). We have explained that the independence between z and a is an inherent property of VRL-PGM and the variational lower-bound (learning objective) we derived based on VRL-PGM from first principle naturally disincentives learning z that carries redundant (dependent) information of a without the need to explicitly enforce it. Again, we would like to stress that we believe the fundamental challenge of relational learning with existing unsupervised learning is not whether or not they can learn relational/absolute property  (in fact, we believe existing methods are adequate for learning both properties) but to learn a relational property that is DECOUPLED from the absolute property (existing methods are not capable of decoupling them). We believe the provided MNIST relational learning example clearly demonstrated this point.
> > >
> > > Lastly, we are grateful for your time and we hope we have adequately address your concerns. If you find the above response useful, we would like to seek your advice on which part you would like us to include in the submission. Please let us know if you have any questions. We look forward to additional discussion.

---

> > > ### Comment · AnonReviewer2 · 2020-11-12
> > > **Responses**
> > >
> > > > We do not fully understand R2’ comment that “example used does not follow the causal model framework” and we look forward to further clarification on the comment and and provide you with a detailed explanation.
> > >
> > > I already explained this in my main review.  "This is causally incorrect because for handwritten digits, the label is the cause of the image, not vice-versa."  However, if it wasn't already clear for you, how do you think MNIST digits are generated?  A human has some address in mind, and then they write glyphs corresponding to those digits, and then those are digitally scanned.  So, do the labels cause the images, or do the images cause the labels?
> > >
> > > > A common practice for reasoning within the latent space is to perform vector arithmetic (think of word2vec in NLP and R2’s suggestion for “subtracting the orientations”). Let us investigate this idea further: after performing the vector arithmetic operation by representing (a,b) as a vector b-a (or a-b) in the learned latent space, we will have 25 unique vectors (defined by starting and end point and include vectors with zero length) that represent all possible combinations of (a,b) (Note that a and b are both random rotations of the same digit and each of them will equally likely be mapped to any of the 5 rotation point in the latent space; thus, we have 5*5=25 different combination between a and b). Finally, we are left with the question on how to infer the rotation relationship (we know there are only 5) from the 25 unique vectors. One may be tempted to categorized these 25 vectors into 5 category based on their features (such as length, direction, etc) but will soon realize that this is not at all a trivial task (this can be intuitively understood by experimenting with placing five point on the paper, construct 25 vectors by making all possible connection with direction, and try to come up with a consistent rule to categorize the 25 different vectors into five category).
> > >
> > > I agree that post-hoc categorization of pairs (z1, z2) would not be promising, but I'm sure that it is possible to learn a rule for transforming z1->z2 and vice-versa with a transformation representation z.  This may not have been proposed before in the literature but it seems quite feasible to me, and a more appropriate model than the one that you propose.

---

> > > > ### Author Response · Authors · 2020-11-12
> > > > **Author response**
> > > >
> > > > Dear Reviewer:
> > > >
> > > > To address your question on the MNIST and its label. Nowhere in our submission did we use the digit label of the MNIST image (except when creating initial test dataset where we generate pair of the SAME digit image but with different rotations). None of the problems and examples we considered involves the prediction/condition on the digit label. In fact, a main goal of our work is learning without the influence of digit label. Throughout our examples, the goal is always to learn/predict relationships and not digit label. The annotation we added to the scatter plot are the "label" for the rotation relationship and not the digits itself. Please allow us to briefly rephrase our problem: 1. We are given a set of (a, b), e.g, a and b are the same MNIST image but with different rotations (note that both a, b are images and not digit labels); 2. We are learning a likelihood function p(b|a,z) where we predict b given z and a (image), and posterior p(z|a,b) where we infer z a posteriori from a, b (again, both are images). We learn z to represent (rotational) relationship and NOT digit label. As you can see, none of the above description involves digit label. We would appreciate if we could get more clarification to your question.
> > > >
> > > > To your second point. It is not possible to provably claim that ALL existing methods (or any combination of existing methods) won't work for relational learning problem (including the provided MNIST example); this is the reason we highly value reviewer and public feedback to show us what we don't know. We can only, to the best of our knowledge, present our rigorous and transparent analysis/examples including our comparison with VAE and InfoGAN, which in our view are the most representative and comparable unsupervised learning methods. We have shared our empirical justification on why existing method (and even the ones R2 have proposed) are fundamentally inadequate for solving the relational learning problem (including the seemly simple MNSIT example). We presented our justification both on a high-level and detailed thought experiment then share our general observations.  We would be more than happy to discuss our response #4, #5 and clarify/discuss any questions your may have, but without further concrete evidence/ examples/counter-examples/references, we respectfully disagree with your assessment that "I'm sure that it is possible to learn a rule for transforming z1->z2 and vice-versa with a transformation representation z" and "it seems quite feasible to me". On the other hand, using your proposed notation, learning the relationship between z1 (derived from a) and z2 (derived from b) IS IN ITSELF a relational learning problem, so we are back to where we started, except now we are working with a reduced representation of a and b. Also, we do not understand your assessment on "This may not have been proposed before in the literature but it seems quite feasible to me, and a more appropriate model than the one that you propose."; more specifically, we cannot comment on future method that has not been proposed before in the literature, let alone compare it with our method.
> > > >
> > > > Based on comments and discussion so far, we believe we have a convincing case that the proposed MNIST relational learning examples, albeit simple at first glance, provides an excellent set of testing cases for future development: it is easy to generate, easy to interpret, and most important of all, exemplifies the key relational learning challenge, so that future methods that "may not have been proposed before in the literature" can be applied on the proposed MNIST relational learning examples and test if  "it is possible to learn a rule for transforming z1->z2 and vice-versa with a transformation representation z".
> > > >
> > > > Thank you for your comment and we look forward to additional discussion.

---

> > > > > ### Comment · AnonReviewer2 · 2020-11-20
> > > > > **Retracting objection about MNIST example; clarification on intent of proposed alternative**
> > > > >
> > > > > > To address your question on the MNIST and its label. Nowhere in our submission did we use the digit label of the MNIST image (except when creating initial test dataset where we generate pair of the SAME digit image but with different rotations). None of the problems and examples we considered involves the prediction/condition on the digit label. In fact, a main goal of our work is learning without the influence of digit label. Throughout our examples, the goal is always to learn/predict relationships and not digit label. The annotation we added to the scatter plot are the "label" for the rotation relationship and not the digits itself. Please allow us to briefly rephrase our problem: 1. We are given a set of (a, b), e.g, a and b are the same MNIST image but with different rotations (note that both a, b are images and not digit labels); 2. We are learning a likelihood function p(b|a,z) where we predict b given z and a (image), and posterior p(z|a,b) where we infer z a posteriori from a, b (again, both are images). We learn z to represent (rotational) relationship and NOT digit label. As you can see, none of the above description involves digit label. We would appreciate if we could get more clarification to your question.
> > > > >
> > > > > OK, thanks.  I now realize that I misunderstood the MNIST example.  I will modify my original review.
> > > > >
> > > > > > On the other hand, using your proposed notation, learning the relationship between z1 (derived from a) and z2 (derived from b) IS IN ITSELF a relational learning problem, so we are back to where we started, except now we are working with a reduced representation of a and b.
> > > > >
> > > > > That is a fair point.
> > > > >
> > > > > > more specifically, we cannot comment on future method that has not been proposed before in the literature, let alone compare it with our method
> > > > >
> > > > > Of course I am not expecting you to implement all possible approaches for solving this problem.  However, I was looking for a justification for why you chose the particular approach you did versus an alternative that seemed more obvious to me.  Indeed, the point you made above, "learning the relationship between z1 (derived from a) and z2 (derived from b) IS IN ITSELF a relational learning problem, so we are back to where we started", is a satisfactory defense of your approach against my alternative.
> > > > >
> > > > > It doesn't impact any of the issues I raised in my original review but it does answer an additional question that occurred to me during the discussion phase.  My thanks to you.

---

> > > > > > ### Author Response · Authors · 2020-11-20
> > > > > > **Thank you, we really enjoyed the discussion and exchanging ideas!!**
> > > > > >
> > > > > > Dear Reviewer:
> > > > > >
> > > > > > We are glad that we can provide additional clarification. We are truly thankful of your time and your interest in our work! We thoroughly enjoyed the discussion and exchanging ideas with you. We hope we don't come off as trying to invalidate your approach, our intention was merely sharing our thought on this idea.
> > > > > >
> > > > > > > However, I was looking for a justification for why you chose the particular approach you did versus an alternative that seemed more obvious to me.
> > > > > >
> > > > > > We do think your proposed alternative approach is very reasonable. In fact, it is similar-in-nature to our earlier attempts at the problem but despite our best effort we couldn't take it very far. However, we still think this is an open question and we are genuinely curious to see how other people would approach this problem !!
> > > > > >
> > > > > > We hope that you find the added experiments satisfactory. Please let us know if you have any unaddressed, or new, question/concern. We are eager to provide additional explanation/clarification!!
> > > > > >
> > > > > > Thank you for reevaluating our work!

---

> > ### Comment · AnonReviewer2 · 2020-11-12
> > **Justification for claims?**
> >
> > > The independence between latent variable z and a is an inherent property of the proposed VRL-PGM and not something we need to enforce explicitly.
> >
> > Why?
> >
> > > therefore, there is no incentive for z to learn redundant (dependent) information from a; this effect is further “enforced” when the learning objective includes additional regularization term on z to learn a compact representation
> >
> > There is no penalty for learning dependence either.  The regularization could help, but could you explain further?
> >
> > > no incentive for z to carry redundant absolute property information from a (only need to carry the decoupled relationship information from a and b)
> >
> > No incentive, but that by itself doesn't enforce independence.
> >
> > > To this end, we would like to present another advantage of the proposed VRL-PGM and the derived variational lower bound: we can add any terms that measure the dependency between z and a (with maximum attained when z and a are independent) to the lower bound (4) WITHOUT invalidating the lower bound since z and a are already assumed to be independent in VRL-PGM! For example, we can safely (without invalidating the lower bound) add the mutual information between z and a, -I(z, a), to Eq. (4) since I(z,a)>=0 and I(z,a)=0 iff z and a are independent (which they are in VRL-PGM!). Please let us know if you would like us add this discussion to the submission.
> >
> > It would be great if you could try this and evaluate whether it makes any difference.

---

> > > ### Author Response · Authors · 2020-11-12
> > > **Author response**
> > >
> > > Dear Reviewer:
> > >
> > > The proposed VRL-PGM belongs to a very specific kind of PGM. A well-known yet counterintuitive property of such PGM is that:
> > > a and z are independent when b is not observed; however, z and a are no longer independent when b is observed! We would refer the reviewer to (Bishop, 2006, Chap. 8.2.1), also referenced in our original submission, for a more complete description. Please also see our response #1 to R1. Moreover, we note that when applying VRL-PGM to the relational learning problem (including MNIST examples), only a is observed/constrained (by data) while z is unobserved/learned/unconstrained so there is absolutely no contradiction to the assumption that z and a are independent.
> > >
> > > Again, we repeat that the independence between z and a  is an inherent property of the proposed VRL-PGM and not something we need to enforce explicitly. Another way to rephrase our explanation in response #2 is that the main learning objective is drive by fitting the likelihood model, p(b|a,z) to fit data b only, and since a is already been conditioned on there is no incentive for z to carry information from a. And this is further enforced/encouraged by other regularization term on z in the learning objective so to encourage a compact representation of z (squeeze out any irrelevant information that does not help with predicting b). While there is no explicit penalty on learning dependency, the independence is naturally encouraged through the interplay between different terms in the learning objective as we have explained. This theoretical property is backed up by our various experimental results where we clearly show that VRL can indeed learned a decoupled (independent) z from the absolute property of a.
> > >
> > > Having explained this, we do understand where your concern is coming from and presented a straightforward way to extend the learning objective to include additional term that enforce/encourage independence between z and a, e.g., negative mutual information. This is an interesting addition indeed! However, we prefer our original learning objective as it is derived from first principle without any heuristics and/or assumptions not provided by PGM. We are happy to try this out but we think the current experimental results (based on the derived learning objective) are already satisfactory and clearly shows that we are learning a decoupled relational property.
> > >
> > > Thank you for your comment and we look forward to additional discussion.

---

> > > > ### Comment · AnonReviewer2 · 2020-11-20
> > > > **Clarification on causal terminology; nontriviality of learning PGMs; objection to claimed purism**
> > > >
> > > > Dear authors:
> > > >
> > > > Thanks for the reference to Bishop.  In fact, I am familiar with the causality literature including the seminal book by Pearl, Causality, as well as several works on related frameworks: potential outcomes, observational studies, inference on experiments with a combination of interventional and observational data (Pfister et al. 2019).  I looked up your reference and I assume you are referring to figure 8.20 in the 2006 edition.  Indeed, the PGM is one of the three equivalence classes for an acyclic causal graph with the skeleton a-b-c.
> > > >
> > > > >  A well-known yet counterintuitive property of such PGM is that: a and z are independent when b is not observed; however, z and a are no longer independent when b is observed!
> > > >
> > > > Your statement is using imprecise (and therefore technically incorrect) language to describe the causal theory illustrated in Bishop (and any other causal inference textbook.). To be more precise, _a_ and _z_ are independent unconditionally.  But conditioning on _b_ , then in the conditional joint distribution of _a_ and _z_, they are potentially dependent (and are dependent in the general case.)
> > > >
> > > > However, you language is incorrect because the dependence or independence has nothing to do with whether or not _b_ is **observed**.  It is entirely a matter of whether you condition on _b_.  What it means to condition on _b_ in the practical data analysis sense is to filter out observations based on a single target value of _b_ or perhaps a subset of possible values.  However, it is evident that this is not what is done in your model.  Hence, it is irrelevant whether or not _b_ is observed in your case--looking at the data distribution, we should see evidence that _a_ and _z_ are independent, otherwise your method is not succeeding at fitting the PGM that you specified.
> > > >
> > > > > Again, we repeat that the independence between z and a is an inherent property of the proposed VRL-PGM and not something we need to enforce explicitly.
> > > >
> > > > On the contrary, your own analysis in your paper shows that you needed to take extra steps in order to make sure that the learned _z_ and _a_ are independent.  Isn't that one of the main motivations behind your section on "Optimization challenges" and the motivation for RPDA?  It goes to show that just because property X holds for a certain PGM, it does not automatically follow that naïvely optimizing the corresponding variational objective will give you a result that preserves property X.  To be more specific, the failure could be due to numerous reasons -- insufficient training data, failure to reach the global minimum, non-identifiability of the model, etc.  This is nontrivial work, and this very paper testifies to that fact.
> > > >
> > > > > However, we prefer our original learning objective as it is derived from first principle without any heuristics and/or assumptions not provided by PGM.
> > > >
> > > > But this didn't prevent you from proposing RPDA... why include RPDA in this paper given that it didn't follow from the original PGM?
> > > >
> > > > Please do not misunderstand me, I am not against RPDA, and I think it adds to the paper.  I am against your hypocritical claim that you refuse to include methods that are not based on variational first principles.  Since you have already shown yourself to be willing to develop methods (again, I am referring to RPDA, and also any kind of regularization you use) for pragmatic reasons, then what is your basis for refusing to try out a mutual information penalty (that you yourself proposed in the first place!)?
> > > >
> > > > Again, just to be absolutely clear.  If adding a mutual information is "impure" in the sense of not being based on variational first principles, then how are data augmentation and regularization, both of which you advocate in the paper, not also "impure"?

---

> > > > > ### Author Response · Authors · 2020-11-20
> > > > > **Thank you for the insightful comments!!**
> > > > >
> > > > > Dear Reviewer:
> > > > >
> > > > > Thank you for the insightful comments! Please allow us to share our thoughts:
> > > > >
> > > > > > To be more precise, a and z are independent unconditionally. But conditioning on b , then in the conditional joint distribution of a and z, they are potentially dependent (and are dependent in the general case.)
> > > > >
> > > > > You are absolutely correct! We recognize that we are sloppy in our original description. We have since made modification to make it more precise in our latest revisions (Section 3.1).
> > > > >
> > > > > > What it means to condition on b in the practical data analysis sense is to filter out observations based on a single target value of b or perhaps a subset of possible values. However, it is evident that this is not what is done in your model.
> > > > >
> > > > > This is accurate. In this work, we are mainly interest in learning about the posterior p(z|a, b) and not the data generating distribution p(a, z).
> > > > >
> > > > > > looking at the data distribution, we should see evidence that a and z are independent, otherwise your method is not succeeding at fitting the PGM that you specified.
> > > > >
> > > > > Thank you for this comment! However, we believe that even when relational property and absolute property are dependent in the data, it is still valid to apply the VRL (but VRL is learning something else!!). We provided one such example (coupled relational property) in appendix B.1.1 where relational property completely (amount of rotation) depends on absolute property (digit representation). In that example, a complete description of the data relationship will be something like: "There are five rotation relationships AND the amount of rotation is determined by the digit representation". When we apply VRL, we discover "There are five rotation relationships. PERIOD." (shown in Fig. 7). Clearly, the decoupled relationship found by VRL did not completely characterize the true relationships but we think it still provide unique and important perspective on the data.  We are very clear about this point and discussed it Section 6 (second paragraph). So yes, when a and z are dependent we are not succeeding at fitting the VRL-PGM to fully characterize the data. But in those situations, VRL still learns important, albeit incomplete, information about the data that we don't know how to learn otherwise. We love to hear your thought on this.
> > > > >
> > > > > > On the contrary, your own analysis in your paper shows that you needed to take extra steps in order to make sure that the learned z and a are independent. Isn't that one of the main motivations behind your section on "Optimization challenges" and the motivation for RPDA?
> > > > >
> > > > > > It goes to show that just because property X holds for a certain PGM, it does not automatically follow that naïvely optimizing the corresponding variational objective will give you a result that preserves property X.
> > > > >
> > > > > > I am against your hypocritical claim that you refuse to include methods that are not based on variational first principles. Since you have already shown yourself to be willing to develop methods (again, I am referring to RPDA, and also any kind of regularization you use) for pragmatic reasons, then what is your basis for refusing to try out a mutual information penalty (that you yourself proposed in the first place!)?
> > > > >
> > > > > > Again, just to be absolutely clear. If adding a mutual information is "impure" in the sense of not being based on variational first principles, then how are data augmentation and regularization, both of which you advocate in the paper, not also "impure"?
> > > > >
> > > > > Fair point! We understood and agree with your viewpoint on regularization and where your initial concern come from;
> > > > > more specifically, the learning objective derived from PGM does not "explicitly" (and actually "discourage") introduce dependency between a and z; however, it also does not "explicitly" guard against it and therefore, cannot be certain it won't happen in practice (even though the learning discourage it). We recognized that this is indeed an important question and warrants further and comprehensive investigation in a future work. Upon reflection, we apologize for appearing overly defensive and to "claim purism".  We will revise our description on this issue in an upcoming revision! Thank you for pointing this out!. As a side note, we would note that our motivation for RPDA is to overcome the information-shortcut problem which is cause by the dependency between z and b introduced by PGM (we added this explanation in D.1).
> > > > >
> > > > > Thank you for sharing your thoughts and feedbacks. We look forward to hear back from you!

---

> ### Author Response · Authors · 2020-11-16
> **Author supplemental response with latest revision**
>
> Dear Reviewer:
>
> We hope that you had a chance to go through our response where we offered detailed explanation to each and every question/concern that was raised in your initial review. In addition to our previous response, we would like to go over your "cons list" again and pinpoint where in the latest revision you'll find a description that addresses your concern (please access our latest revision):
>
> > toy examples inadequate to show novelty and generalizability of the method
>
> A detailed explanation on why the presented MNIST relational learning experiments are relevant and significant (challenging for existing methods) is added in Section 5.1.
>
> > one toy example does not fit model assumptions (seemingly assuming that MNIST images cause labels rather than vice-versa)
>
> We hope our response #3 and subsequent discussion fully answers these questions.
>
> > not obvious to me how the objective function encourages independence of z and a.
>
> We hope our response #2 and subsequent discussion fully answers these questions. Additional explanation on VRL-PGM is provided in appendix D.1 (a pointer is added in the main text).
>
> > do we have any confidence that this method would work in a more complicated problem?
>
> Additional real-world relational learning experiments with face dataset that involves complex, high-level perception reasoning is provided in appendix B.2. Additional justification on why we believe the proposed method is robust, stable, and generalizable is provided in appendix D.3.
>
> We hope that you find these additions satisfactory and consider our work favorably. Please let us know if you have any questions. We look forward to additional discussions.

---

> ### Author Response · Authors · 2020-11-21
> **Latest revision updates discussion on independence of z and a**
>
> Dear Reviewer,
>
> We have updated our submission to reflect our discussion on the independence of z and a.
> More specifically, we include a discussion in appendix D.2 that explains how, in an ideal setting,
> optimizing the variational learning objective would encourages learning independent z and a; and in
> a practice, we can extent the learning objective to  explicitly safeguard against
> introducing dependency between z and a. A pointer is added in the main text at the end of Section 3.2.
>
> We hope this addresses your (and other readers) concern on how the objective function encourages independence of z and a.
> Please let us know if you have any questions.
>
> Thank you,

---

### Official Review · AnonReviewer3 · 2020-10-28

**Rating:** 6
**Confidence:** 3

**Review:**

The paper proposes variational relational learning by learning relations between two inputs via variational inference on a probabilistic graphical model (PGM). The PGM that they use factors as p(a)p(z)p(b|a,z) where a,b are the two inputs and z is the supposed relationship between them. The example shown in the experiments is rotational mnist, where b is a rotated version of a, and z should encode the degree of rotation. The paper learns both the forward network and in inference network in a VAE-like approach; the elbo derivations appear correct to me.

In section 3.3 the paper describes two sources of problems with naively optimizing based on the above approach: 1) deterministic mapping where a completely determines b, and no learning of relation is necessary, and 2) information shortcut where z can completely encode b. To me 1 does not seem like a big drawback since it is more of a limitation with the underlying data. For 2, the paper describes a data-augmentation technique they call RPDA, as well as using an informative prior on z. The experimental results seem reasonable, although having setting beyond mnist would have been nice.

The two methods described to handle the information shortcut issue are not super satisfactory in my opinion. The data augmentation technique relies on designing augmentations g that preserve the relationship, so that (a,b) are related in the same way as (g(a), g(b)). But coming up with such augmentations seems to rely on us knowing properties of the relationship z, which is exactly what we are trying to learn in the first place. The alternative of using an informative prior is not super clear to me, since it seems that with a powerful enough model p(b|a,z), scaling and shifting p(z) to fit a Gaussian will not prevent the model from completely encoding b as a function of z.

In my opinion constraining the expressiveness of z seems like the right path, but instead of regularizing p(z) to be a Gaussian, I would consider using discrete z's that explicitly limits the theoretical # of bits z can store, hence preventing it from encoding b. I think this also intuitively aligns with how many relations seem more discrete than continuous (a is friend of b, a is adjective form of b, ...etc). I would be interested in seeing the experiments done using discrete z's, which seems natural too given the 5 discrete rotations considered in the paper.

---

> ### Author Response · Authors · 2020-11-12
> **Author response to Reviewer 3 review**
>
> Dear Reviewer,
>
> We would first like to express our deep appreciation for your time and insightful comments. Please find our response to your concerns (referring to as R3) in the order they are raised:
>
> 1.	We believe the provided MNIST relational learning examples represent novel and unique questions and clearly demonstrate the relational learning challenge that are not easily solvable with existing methods (please see a detailed example and explanation in our response #4, #5 to reviewer R2).  In addition to the MNIST example presented in the main text, we include three additional MNIST relational learning example in the appendix each with increasing complexity (coupled relational property, multiple relational property, and continuous relational property). We demonstrate the robustness of the proposed method by successfully achieving satisfactory results on all four problems with the exact same model setup. To further demonstrate the generalization ability of the proposed method on a more complicated problem, we presented a face relational learning example and successfully solved it with a very similar setup as in our MNSIT example where we only made necessary adjustment to the model setup to accommodate the different types of data (e.g., larger image and continuous vs binary data).
>
> 2.	We think R3’s suggestion on “constraining the expressiveness of z” nicely sums up what we intended to say in our original submission. We think imposing an informative prior with restrictive constraint also falls under the strategy of “constraining the expressiveness of z”. We also agrees that a discrete prior is appropriate for learning a set of discrete relational property (when they are known in advance and appropriate number of set is chosen!). With R3’s permission, we would like to go ahead with changing the wording in our submission as well as including the discussion of discrete prior. We hope to convince the readers that our main contribution is the proposal of a general and flexible variational learning framework for relational learning that can applied in conjunction with different prior based on the understanding/assumption of the problem. While we experiment with this idea, our original motivation for choosing a Gaussian (continuous) prior is to have a consistent model setup to test all four different relational learning scenarios: coupled relational property, decoupled relational property, multiple relational property, and continuous relational property. Our goal, beyond the fact that we want to strictly restrict our self to “completely unsupervised” setting where we don’t exploit the knowledge of knowing the under relational property to be discrete, is to demonstrate the robustness (not sensitive to model setup) and generalization of the proposed method.
>
> Lastly, we are grateful for your time and we hope we have adequately address your concerns. If you find the above response useful, we would like to seek your advice on which part you would like us to include in the submission. Please let us know if you have any questions. We look forward to additional discussion.

---

> ### Author Response · Authors · 2020-11-16
> **Author supplemental response with latest revision**
>
> Dear Reviewer:
>
> We hope that you had a chance to go through our response where we offered detailed explanation to each and every question/concern that was raised in your initial review. In addition to our previous response, we would like to go over your "concern list" again and pinpoint where in the latest revision you'll find a description that addresses your concern (please access our latest revision):
>
> > The experimental results seem reasonable, although having setting beyond mnist would have been nice.
>
> Additional real-world relational learning experiments with face dataset that involves complex, high-level perception reasoning is provided in appendix B.2.
>
> > coming up with such augmentations (RPDA) seems to rely on us knowing properties of the relationship z, which is exactly what we are trying to learn in the first place.
>
> A detailed discussion on the practical applicability of RPDA is provided in appendix D.2 (a pointer is added in the main text).
>
> > the expressiveness of z seems like the right path...
>
> We agree and have modified the main text in Section 3.3.
>
> > I think this also intuitively aligns with how many relations seem more discrete than continuous (a is friend of b, a is adjective form of b, ...etc).
>
> We agree this is reasonable and interesting. We would remark that our original intent was to use the exact same model and training setup for all of our experiments (including a continuous relationship experiment in appendix B.1.3) to demonstrate the proposed method is robust, stable, and generalizable  (discussion provided in appendix D.3).
>
> We hope that you find these additions satisfactory and consider our work favorably. Please let us know if you have any questions. We look forward to additional discussions.

---

> > ### Comment · AnonReviewer3 · 2020-11-16
> > **Response**
> >
> > Thanks for your response, I appreciate the revisions.
> >
> > I still have an issue with relying on augmentation techniques to learn relationships.
> >
> > From Appendix D.2: "the construction of RPDA functions D reflects our prior knowledge and belief of the underlying system and not based on the underlying relational property of the data". But what is the difference here, really? In the rotational MNIST example, the whole point is to learn the different images are related by rotations. If you are already injecting rotational invariance, then why is a learning approach even necessary? You could take every image, rotate it, and search over the dataset for a match.  It seems that this will just learn whatever prior invariance you assert (e.g. if you assert that the middle horizontal pixels can be flipped, then 0's will be related to 8's). So my dissatisfaction is that: the approach will just spit out whatever invariance relationship we asserted (and if we already had access to this invariance relationship, what is the problem we're really solving?).

---

> > > ### Author Response · Authors · 2020-11-16
> > > **Author response**
> > >
> > > Dear Reviewer:
> > >
> > > Thank you for the question!! This is a key point so please allow us to further clarify. The key to answer your question is to see that we treat (a,b) as ONE inseparable data point (e.g., think of a and b are the different channels of a multi-channel (colored) image). So when we say we want the learning to be "rotation invariance", what we mean is that we want the learned model to be indifferent to "rotation" of the ENTIRE inseparable data point (a,b) (e.g., rotation of a colored image). In our MNIST rotational example, the RPDA applies the SAME random rotation to BOTH a and b (as one inseparable data point) at the SAME time; in other words, we are "injecting rotational invariance" to (a,b) as one inseparable unit and this clearly does not change the relationship BETWEEN (a,b). What we hope to convey in the paper (discussed in length in Section 3.3 and appendix D.2) is that RPDA only sees and treats (a,b) as ONE inseparable data point and does not care nor modify the relationship BETWEEN (a,b). Put it in another way, RPDA applies data augmentation to the ENTIRE (a,b) while the problem objective is to learn relationship BETWEEN (a,b). We discussed in appendix D.2 that in many instances this can be done WITHOUT any knowledge or assumption on the underlying data relationships.
> > >
> > > > So my dissatisfaction is that: the approach will just spit out whatever invariance relationship we asserted (and if we already had access to this invariance relationship, what is the problem we're really solving?).
> > >
> > > All the experiments presented in the paper (including appendix) are solved with the SAME rotation RPDA despite each experiment involves a very different kind of relationships. For all of the experiments, VRL with the SAME rotation RPDA successfully learns the underlying relationship such as rotation, scaling, high-level perception reasoning (emotion, illumination condition which are clearly very different from image rotations).
> > >
> > > Thank you again for your attention and taking an interest in our work! Please let us know if this explanation clarifies your question. We look forward to provide you with additional explanation/example.

---

### Official Review · AnonReviewer1 · 2020-10-28
**Review of "Relational Learning with Variational Bayes"**

**Rating:** 6
**Confidence:** 3

**Review:**

This paper proposes a model to infer the relationship between multiple instances in a dataset by inferring a latent variable. The authors accomplish this by defining an optimization problem that optimizes the ELBO of the proposed graphical model. The paper presents a nice solution to some of the identification issues that can arise when inferring the latent variable, in particular the so called “information shortcut” when the model overfits to only learning the “absolute” property of the dataset, rather than inferring the shared latent traits.

I think that the formulation that the authors present and the proposed  optimization model is quite interesting, however I have additional questions / concerns:

It’s not clear exactly what the authors mean by a relationship.
Why is there a latent variable causing b but not a? Why are we assuming that a causes b?
It’s unclear to me why we refer to the relational property as ‘z’. According to figure 1 z has no effect on a, so it doesn’t seem like it would describe the relationship between a and b? Wouldn’t theta be serving that purpose?

I’m also confused about the difference between what is proposed here and a latent factor model. It would appear that the authors are proposing a model to integrate a latent factor model into a generative net, which is interesting, but does not come through in the text as it currently reads. It would seem that in order to interpret z as a relational variable we should be able to extract some kind of meaning from it? Also, if there are multiple relationships , say “animal”, “rotation”, and “saturation”, should we expect to be able to disentangle these concepts with the proposed model?

I found the experiments to be a bit underwhelming. It is unclear how the authors decided on the architecture, hyperparameters, and number of latent variables for the proposed model. It would seem that the model would be quite sensitive to these. In addition, it seems that some of these evaluations would benefit from comparison to more traditional methods. For example, the faces example appears in “A Dependence Maximization View of Clustering”, within a very similar context.

Overall, I think this is an interesting idea, but I would like to see the paper a bit more refined before recommending acceptance.

---

> ### Author Response · Authors · 2020-11-12
> **Author response to Reviewer 1 review**
>
> Dear Reviewer:
>
> We would first like to express our deep appreciation for your time and insightful comments. Please find our response to your concerns (referring to as R1) in the order they are raised:
>
> 1.	We interpret the first questions/concerns as stemming from our overly succinct introduction and explanation of the proposed VRL-PGM and its connection to the original relational learning problem. Please allow us to clarify: first,  we interpret the relationship (relational property) as any information governing the variation between the data sample (a-b) while absolute property as any information governing the features within the data sample (a or b). The goal of relational learning is to discover a set of relational property that is independent (decoupled) from the absolute property. With this goal in mind, we proposed our VRL-PGM model for representing the abstract relational learning problem. The random variables in our VRL-PGM represent the three key information in the original problem: the observed random variables representing a,b and the latent variable z representing the unobserved relational property. We connected a, b, and z with a directed acyclic graph (as shown in VRL-PGM) that reflects our priority as well as compromises for adopting a rigorous PGM to the original abstract problem: our first and foremost priority is to learn a decoupled relational property and this is achieved by the inherit property of VRL-PGM where z is independent of a. In VRL-PGM, the relational property, as represented by latent variable z, can be interpreted as any additional information not found in a but can help to better predict b; in other words, VRL-PGM interpret z (relational property) as an external factor (not found in a) that dictates the relationship (represented by the conditional distribution p(b|a,z)) between a and b. Regarding the specific question on "Why is there a latent variable causing b but not a?", the answer is that we are not interested in learning about a and the fact that b is caused by BOTH a and z can be interpreted as b derives its absolute property from a and relationship information from z (the absolute property of a alone is not enough to characterize b, we need additional relationship information from z). Regarding the specific question on "According to figure 1 z has no effect on a, so it doesn’t seem like it would describe the relationship between a and b?", the key to answer this question is to recognize the fact that in the proposed VRL-PGM z and a are independent (R1 correctly stated this in the first part of the question); however, they are no longer independent when b is observed (a well-known yet counterintuitive property, see (Bishop, 2006, Chap. 8.2.1))! This means that it is valid to inquire about the posterior p(z|a,b) and it is in this sense that z captures the relationship information between a and b!. Next, we note that there are at least two compromises made by VRL-PGM: first, the original problem specifies that the relational property be decoupled from both a’s and b’s absolute properties; however, the latent variable z that is used to represent relational property in VRL-PGM is only independent of a but not b. Second, we note that the original problem is inherently undirected with no cause-effect relationship between a and b, whereas VRL-PGM is based a directed acyclic graph that artificially introduces conditional dependency between a and b. However, we argue that the application of VRL does not require the true conditional dependency between (a; b) be known in advance only that it is maintained consistently throughout learning and inference, i.e., VRL can be applied in the same way to learn about the relational property between (b; a), where we swap a and b. In short, the artificial causal relationship introduced by VRL-PGM reflects our priority and compromise for representing the original abstract relational learning problem with a PGM: we sacrificed some identifiability of the original abstract problem but gained a rigorous and mathematical tractable statistical model (reasoning and learning in a directed graph is much easier compared to undirected graph) while achieving our primary objective of learning an independent (decoupled) relational property. We hope this discussion clarify R1’s questions regarding the meaning of z as well as its relationship to a and b. We have added this important discussion in the appendix.
> 2.	We think the proposed VRL-PGM is a form of latent (factor) variable model where we designate the latent variable z to represent the relational property and derive a variational inference algorithm that estimate z from a and b. We also estimate the likelihood function p(b|z, a) as a part of the VI learning process. One can certainly view p(b|z,a) as a data generating function for b given a and z, but we do not (at least not the focus of this work) learn p(b|a) or p(b, a).
>
> (To be continued in the appending post)

---

> > ### Author Response · Authors · 2020-11-12
> > **Author response to Reviewer 1 review (Cont.)**
> >
> > 3.	Reviewer 1 raised an interesting question regarding what happens when there are multiple relationships? We presented one such example in the appendix of our original submission: we included a MNIST relational learning example where there are both scaling (by x1 and x1.5) and rotation effect. What we find (and presented in detail) is that our proposed method identified each combination of scaling and rotation as a unique relationship. We believe this observation should also carry over to other multiple relationship example. On the other hand, it is an interesting question to ask whether we can further disentangle the relationships. Our initial thinking is that we  believe our proposed VRL framework provide a basis that can be built upon to answer this question by further constraining (regularizing) the latent variable z, e.g., apply the existing work on unsupervised disentanglement on variable z. This will be an interesting research question in our future work!
> > 4.	We acknowledge that the presented experimented may seem contrived at first glance but we do believe the provided MNIST relational learning examples represent novel and unique questions and clearly demonstrate the relational learning challenge that are not easily solvable with existing methods (please see a detailed discussion in our response #4, #5 to R2). Please allow us to elaborate on our experiment design. To the best of our knowledge, we believe relational learning represent a relatively new (overlooked) problem in this community especially as an unsupervised learning problem. Our main goal for experimental design are: to present a well-controlled problem that clearly demonstrate the key challenges of the relational learning (especially for existing methods); an easily reproducible dataset that are easy to generate and experiment on. We believe the experiment presented in this work (including appendix) achieved both of these goals. Based on the same MNIST data settings we presented 4 different relational learning scenario: coupled relational property, decoupled relational property, multiple relational property, and continuous relational property, each with increasing complexity. Furthermore, to the best of our knowledge, we believe none of these seemly simple examples can be easily solved by existing unsupervised methods and represent a key challenge for existing approach to the relational learning problem (please see a detailed example and explanation in our response #4, #5 to R2). On top of these examples, we also supplemented a more complicated problem with face dataset to test the capability of our method.
> > 5.	To address reviewer R1’s concern regarding model sensitivity and generalization in our experiment: we deliberately design all of our MNIST relational learning experiments using the same model setup despite each experiment represent a very different relational learning scenario (discrete vs. continuous, coupled vs. decoupled). Moreover, the neural net used in our experiment are based on conventional NN setup for MNIST and we adopted common optimizer with widely used hyperparameters. Our experiment results shows that the same (and commonly used) model setup is able to achieve satisfactory relational learning results across all four problem setting and therefore we believe our proposed method is not sensitive to the design of the neural net and are robust to generalization. Regarding the selection of latent space dimension (number of latent variables), we choose 2 for all of our experiment for ease of visualization (presentation) purpose (difficult to visualize beyond 3, and even 3 is difficult to present on paper!). In our continuous relational property experiment, we demonstrated that 2D latent space is not adequate to embed the continuous relationship information (but VRL still learned a reasonable embedding) and we present results with 3D latent space. We did adjusted the model setup for the face example but these modification are only made to accommodate the different types of data (e.g., larger image and continuous vs binary data) and not based on the underlying relationships. Ultimately, we viewed (and hoped to convince the readers) our main contribution as the proposal of a general and flexible variational learning framework for relational learning such that any parametric representation or function approximation (neural net in our example) for the likehood as well as posterior can be easily adopted (plug-and-play).
> >
> > (To be continued in the appending post)

---

> > > ### Author Response · Authors · 2020-11-12
> > > **Author response to Reviewer 1 review (Cont.)**
> > >
> > > 6.	We thank R1 for pointing us to the work of  “A Dependence Maximization View of Clustering”. Upon reading the work, we would like to comment on the similarity and difference between our method and theirs, and share our general observation. Both methods cluster data in an unsupervised manner (we do not view our method as a clustering method nor did we claim it in the paper, the natural clustering appears in the experiment is not guaranteed by the current proposed method; however, it is an interesting research direction warrant further investigattion; our initial thought is that  we believe clustering may be achieved by imposing different prior assumption in our proposed method). Despite the different ways of learning the cluster (they solve an optimization problem and we used amortized variational inference), a fundamental difference (and the key difference in our view!) is that they (also the majority, if not all, of existing unsupervised learning method) cluster data based on both absolute property and relational property (we believe current method are heavily biased towards absolutely property) while we cluster data on a decoupled relational property. This point is apparent in their experimental results where they cluster face images based on identity as well as emotions for individual images. In our results, also the goal of relational learning, we explicitly want to disregard the identity and individual emotion information and solely cluster data by the “emotion changes” between a paired face images (we view identity and emotion as absolute property that describe individual image and “emotion change” as relational property). As can be seen from our results, we learned a clustering structure that have face images with different identity as well as individual emotions group together that share the same emotion change, e.g., happy-sad. In this regard, we do not think their work is comparable to ours (we have fundamentally different goals). For completeness, we did include comparison of our method to VAE on MNIST dataset. VAE is widely understood as capable of learning a latent representation that group similar data together but as we have shown in the experiment, the similarity between data samples learned by VAE are heavily based on absolutely property. Our observation, as detailed in the original submission, essentially highlight the key points in the above discussion.
> > >
> > > Lastly, we are grateful for your time and we hope we have adequately address your concerns. If you find the above response useful, we would like to seek your advice on which part you would like us to include in the submission. Please let us know if you have any questions. We look forward to additional discussion.

---

> ### Author Response · Authors · 2020-11-16
> **Author supplemental response with latest revision**
>
> Dear Reviewer:
>
> We hope that you had a chance to go through our response where we offered detailed explanation to each and every question/concern that was raised in your initial review. In addition to our previous response, we would like to go over your "concern list" again and pinpoint where in the latest revision you'll find a description that addresses your concern (please access our latest revision):
>
> > It’s not clear exactly what the authors mean by a relationship ...
>
> We hope our response #1 fully answers these questions. A detailed explanation on VRL-PGM and its connection to the relational learning problem is provided in appendix D.1 (a pointer is added in the main text).
>
> > I’m also confused about the difference between what is proposed here and a latent factor model ...
>
> We hope our response #2, #3 fully answers these questions. A short discussion on VRL and generative model (generative net) is provided in Section 6. A MNIST relational learning experiment with multiple relationship is provided in appendix B.1.3.
>
> > I found the experiments to be a bit underwhelming.
>
> A detailed explanation on why the presented MNIST relational learning experiments are relevant and significant (challenging for existing methods) is added in Section 5.1. Additional real-world relational learning experiments with face dataset that involves complex, high-level perception reasoning is provided in appendix B.2. Additional justification on why we believe the proposed method is robust, stable, and generalizable is provided in appendix D.4.
>
> >  comparison to more traditional methods. For example, the faces example appears in “A Dependence Maximization View of Clustering”
>
> We have included the referenced paper in the bibliography and explain why it is not comparable at the end of appendix B.2.2. A comparison with VAE and InfoGAN (which in our view are the most representative and comparable unsupervised learning methods) is provided in Section 5.3.
>
> We hope that you find these additions satisfactory and consider our work favorably. Please let us know if you have any questions. We look forward to additional discussions.

---

> ### Comment · Area_Chair1 · 2020-11-18
> **Author response**
>
> Dear AnonReviewer1,
>
> We are now entering the second discussion stage. Could you please check whether the authors have addressed your concerns and questions and potentially ask any further clarification questions?
>
> Thank you,
> Your Area Chair

---

> ### Author Response · Authors · 2020-11-22
> **Nearing end of review; Call for questions; Kindly ask for reevaluation**
>
> Dear AnonReviewer1:
>
> We are near the end of stage-2 review and we have made every effort to address your initial question/concern. In our response below you can find pointers to our latest revision that address each and every one of your question/concern; or you can find a detailed answer in our first response. We are confident that our response/revision should address all your question/concern and we are here to answer any additional ones you may have. If you do find our response/revision satisfactory and have no additional question/concern, may we kindly ask you to take our revised submission into consideration and reevaluate your score?
>
> Thank you again for your time and consideration!!
>
> Authors

---

### Official Review · AnonReviewer4 · 2020-11-01
**An interesting variational framework for relational learning**

**Rating:** 5
**Confidence:** 3

**Review:**

In this paper, the author proposed a variational framework for relational learning that decouples relational property and absolute property among objects based on a pre-defined probabilistic graphical model (PGM). The author also proposed the so called relation-preserving data augmentation (RPDA) strategy to address the challenges for the resulting optimization. Overall, the paper is well written and easy to follow. Below are some of my concerns.

1. It seems that RPDA is crucial for training. However, the relation preserving function D is often difficult to find for real data where the existing relation patterns are often unknown.

2. No ablation study is reported regarding the effect of RPDA.

3. Experiments seem to be too simple. It might be hard to find appropriate application cases though.

4. Can the author clarify the relation between the proposed PGM based relation learning framework and causal learning?

---

> ### Author Response · Authors · 2020-11-12
> **Author response to Reviewer 4 review**
>
> Dear Reviewer:
>
> We would first like to express our deep appreciation for your time and insightful comments. Please find our response to your concerns (referring to as R4) in the order they are raised:
>
> 1.	We acknowledge R4’s comments that in practice relation preserving function D is not always possible to identify in the presence of unknown relationship --- a remark we have expressed in our original submission. However, our viewpoint is that in many practical problem settings, relation preserving function D can be designed without any knowledge of the underlying relational property. For example, in computer vision applications, if we want the learned model to be rotation invariant (a common practice) we can use image rotations as D. Another example may be: in many spectral imaging problem, the orientation of images is not preserved or not enforced (only that they are consistent between the same paired images) and in such problem settings it is safe to adopt image rotations as D. Yet, another example may be: for a discrete time-series data $\mathbf{a}[t], \mathbf{b}[t]$ that represent the input and output of a linear time-invariant (LTI) system (commonly assumed in signal processing and control theory), and we want to learn a relational property that characterize the system's impulse response. We have $\alpha \mathbf{b}[t-\tau]=\alpha \mathbf{a}[t-\tau],\;\forall \alpha \in \mathbb{R}, \tau \in \mathbb{Z}$, and we can construct $D$ with $d(\mathbf{a}[t], \mathbf{b}[t]; \alpha, \tau) = (\alpha \mathbf{a}[t-\tau], \alpha \mathbf{b}[t-\tau])$, $\alpha,\tau \in R=\mathbb{R}\times\mathbb{Z}$. In all of the above example, the relation preserving function D is selected not based on the underlying relational property of the data but the nature of the problem and constraint we impose on our model; therefore, in many instances, RPDA can be designed without any knowledge of the underlying relational property.  The bottom line is that RPDA is not central to the theory of the proposed method (one can apply the proposed VRL method without RPDA) but rather a practical data augmentation strategy for addressing the unique optimization challenge of VRL learning. In the experiments presented in our submission, we find that RPDA to be very effective in overcoming the information-shortcut problem but, like any data augmentations, this is problem dependent and we advocate to start without RPDA and only apply it when necessary (when suspecting information-shortcut occurs).
>
> 2.	We have conducted a detailed ablation study on RPDA in our original submission. Due to the page constraint, we include those results in the appendix.
>
> 3.	We acknowledge that the presented experimented may seem contrived at first glance. However, we do believe that the provided MNIST relational learning examples represent novel and unique questions and clearly demonstrate the relational learning challenge that are not easily solvable with existing methods.  In addition to the MNIST example presented in the main text, we include three additional MNIST relational learning example in the appendix each with increasing complexity (coupled relational property, multiple relational property, and continuous relational property). We demonstrate the robustness of the proposed method by successfully achieving satisfactory results on all four problems with the exact same model setup. To further demonstrate the generalization ability of the proposed method on a more complicated problem, we presented a face relational learning example and successfully solved it with a very similar setup as in our MNSIT example where we only made necessary adjustment to the model setup to accommodate the different types of data (e.g., larger image and continuous vs binary data). Please see a detailed discussion in our response #4, #5 to R2.
>
> (To be continued in the appending post)

---

> > ### Author Response · Authors · 2020-11-12
> > **Author response to Reviewer 4 review (Cont.)**
> >
> > 4. If we understand R4’s comments correctly, we can certainly view our proposed VRL-PGM in the context of causal learning. We would note that the causal relationship that we introduce (in the form of directed acyclic graph) represent our unique interpretation to the abstract relational learning problem and the primary motivation for our work is to utilize the independence property of the proposed PGM (we have added more detailed discussion in appendix that further explains VRL-PGM and its connection to the original abstract relational learning problem). As in causal learning, we strive to learn a model that explains the data; however, the unique aspect of our work is that we are learning a relational property that explains the data, and at the same time, requiring it to be decoupled from the absolute property. In this regard, we believe we are achieving a more targeted causal learning goal (learning decoupled relational and absolute property)
> >
> > Lastly, we are grateful for your time and we hope we have adequately address your concerns. If you find the above response useful, we would like to seek your advice on which part you would like us to include in the submission. Please let us know if you have any questions. We look forward to additional discussion.

---

> ### Author Response · Authors · 2020-11-16
> **Author supplemental response with latest revision**
>
> Dear Reviewer:
>
> We hope that you had a chance to go through our response where we offered detailed explanation to each and every question/concern that was raised in your initial review. In addition to our previous response, we would like to go over your "concern list" again and pinpoint where in the latest revision you'll  find a description that addresses your concern (please access our latest revision):
>
> > 1. It seems that RPDA is crucial for training. However, the relation preserving function D is often difficult to find for real data where the existing relation patterns are often unknown.
>
> A detailed discussion on the practical applicability of RPDA  is provided in appendix D.3 (a pointer is added in the main text). In short, in many instances RPDA can be designed without any prior knowledge of the data relationships.
>
> > 2. No ablation study is reported regarding the effect of RPDA.
>
> A detailed ablation study is provided in appendix C (a pointer is added in the main text).
>
> > 3. Experiments seem to be too simple. It might be hard to find appropriate application cases though.
>
> A detailed explanation on why the presented MNIST relational learning experiments are relevant and significant (challenging for existing methods) is added in Section 5.1. Additional real-world relational learning experiments with face dataset that involves complex, high-level perception reasoning is provided in appendix B.2. Additional justification on why we believe the proposed method is robust, stable, and generalizable is provided in appendix D.4.
>
> > 4. Can the author clarify the relation between the proposed PGM based relation learning framework and causal learning?
>
> We hope our response #4 fully answers this question.
>
> We hope that you find these additions satisfactory and consider our work favorably. Please let us know if you have any questions. We look forward to additional discussions.

---

> > ### Comment · AnonReviewer4 · 2020-11-24
> > **Response**
> >
> > Thanks for your response, I appreciate the revisions.
> >
> > I still have doubt about the usage of certain RPDA for unknown relational learning. For that, I think I agree with Reviewer 3. Given that RPDA is crucial in the learning process (validated by the ablation study in appendix C), and the experiments are relatively limited  in terms of the diversity of relationships to be learned, I am not fully convinced that the proposed approach would be generally applicable and hence keep my score unchanged.

---

> > > ### Author Response · Authors · 2020-11-24
> > > **Author response to general applicability of RPDA**
> > >
> > > Dear AnonReviewer4:
> > >
> > > We appreciate you sharing your concern at the last minute. Regardless of your final score we would like to address
> > > your concerns regarding RPDA:
> > >
> > > > I still have doubt about the usage of certain RPDA for unknown relational learning. For that, I think I agree with Reviewer 3
> > >
> > > Based on our full discussion with Reviewer 3, we believe we have fully and satisfactorily addressed Reviewer 3's concerns regarding both the practical applicability of RPDA as well as its mechanics. Again, we have provided specific examples in appendix D.3 that shows the general applicability of RPDA in many common problem settings without any prior knowledge of underlying data relationship.
> > >
> > > > Given that RPDA is crucial in the learning process (validated by the ablation study in appendix C), and the experiments are relatively limited in terms of the diversity of relationships to be learned, I am not fully convinced that the proposed approach would be generally applicable.
> > >
> > > We have presented a wide range of experiments that have all been applied with the SAME rotation RPDA. With these results, we believe we have successfully demonstrated, to the extent that a paper allows, the following three things: 1. RPDA is EFFECTIVE, as shown in our ablation study; 2. RPDA can be used with common (non-controversial) image rotation data augmentations for learning a diverse set of data relationships (rotational relationships, scaling relationships, continuous relationships, high-level perception relationships, etc.); 3. RPDA can be applied WITHOUT any knowledge of the underlying data relationships (whether of not the underlying data relationship is similar to the RPDA function). On a higher level, we believe our answer to your question regarding RPDA's general applicability is entirely consistent with the general applicability of ANY data augmentation (such as the widely used image rotation augmentation): just like ANY data augmentation, it's applicability will certainly depend on the specific learning task; nevertheless it can be easily adopted (leveraging existing experience with data augmentation) and highly effective in preventing overfitting (or providing model regularization). The bottom line is that, if the user is comfortable with the general concept of data augmentation, he/she should also be comfortable with RPDA (because both of them have the same requirement and applicability).
> > >
> > > Next, we want to re-emphasize that (also clearly stated in our submission) the KEY and NOVEL contribution of RPDA is not the specific data augmentation function (e.g. rotation) we used in the experiments, but our development of Eq. 7 which tells you EXACTLY which term in the learning objective TO APPLY data augmentation and which term NOT TO APPLY. In this sense, RPDA is a general and flexible data augmentation STRATEGY that can be used with any existing (what's already been used in practice) data augmentation functions. Hence, we conclude our ablation study by saying "the key contribution of RPDA comes not from what data augmentation functions are applied but how they are applied"
> > >
> > > Finally, as explained in our submission, RPDA is developed to overcome a specific and unique optimization challenge---information-shortcut---of VRL learning. We presented two ways for overcoming this challenge: constraining the expressiveness of the latent variable z, and RPDA. If constraining z is more suitable (e.g., impose discrete prior on z), or suspect information-shortcut did not occur, then there is no need to apply RPDA. Again, this principle of RPDA application (agnostic to the learning task and apply only when needed) is entirely consistent with the application of any data augmentation.
> > >
> > > We hope you will have a chance to review our response.
> > >
> > > Thank you,
> > >
> > > Authors

---

> > > > ### Comment · AnonReviewer4 · 2020-11-25
> > > > **Response**
> > > >
> > > > If RPDA does not rely on the relation to be learned (which seems like cheating already), can the author try other data augmentation methods (e.g., adding Gaussian noise) in RPDA on the same MNIST data set and report the result?

---

> ### Comment · Area_Chair1 · 2020-11-18
> **Author response**
>
> Dear AnonReviewer4,
>
> We are now entering the second discussion stage. Could you please check whether the authors have addressed your concerns and questions and potentially ask any further clarification questions?
>
> Thank you,
> Your Area Chair

---

> ### Author Response · Authors · 2020-11-22
> **Nearing end of review; Call for questions; Kindly ask for reevaluation**
>
> Dear AnonReviewer4:
>
> We are near the end of stage-2 review and we have made every effort to address your initial question/concern.
> In our response below you can find pointers to our latest revision that address each and every one of your
> question/concern; or you can find a detailed answer in our first response. We are confident that our
> response/revision should address all your question/concern and we are here to answer any additional ones you may have.
> If you do find our response/revision satisfactory and have no additional question/concern, may we kindly ask you to
> take our revised submission into consideration and reevaluate your score?
>
> Thank you again for your time and consideration!!
>
> Authors

---

### Author Response · Authors · 2020-11-13
**Update on submission with new complex experimental results added**

Dear reviewers and readers:

Based on the continuous feedbacks, we have made several important updates to our submission:

1. We have added "Additional remarks" section in the appendix that include clarification and additional explanation on VRL-PGM, RPDA, and experimental results (compiled from our discussion).

2. As per reviewer's suggestion to add a more complicated example, we have added a new relational learning experiment for learning "illumination condition change" among Extended Yale Face Database B in the appendix. Together with the “learning emotional changes" included in our original submission, these two examples represent a real-world relational learning problem that involves complex, high-level perception relationships (sentiment, external environmental factors) reasoning.

We hope you will find these additional material satisfactory and consider our work favorably. Please let us know if you have any questions. We look forward to additional discussion.

---

### Author Response · Authors · 2020-11-17
**Author summary of stage-1 review**

Dear Reviewers, Readers, AC:

We are the end of stage-1 review and I wanted to thank you again for your time and feedbacks. Based on all the questions, feedbacks, and discussions, we would like to share our summary of the review so far. We will present the summary along three topics: problem, method, and experiment.

1.	Problem: Relational learning is an important and long standing problem; however, it is a relative new problem for the machine learning community (especially for unsupervised learning community). We hope this work will draw community’s attention to this important problem.

2.	Method: The main contribution of this work is the proposal of VRL method, which we believe is one of the first unsupervised learning method for relational learning. VRL is developed based on a rigorous and tractable VRL-PGM but made two compromises to the original problem that leads to unique learning challenges. We analyzed these challenge and proposed a novel RPDA strategy for overcoming them. We hope our response and added material resolve all of reviewer’s concern; so far we do not believe there are outstanding questions about the novelty, validity, and technical correctness of the proposed VRL and RPDA.

3.	Experiments: We presented 4 MNIST experiments that covers a wide variety of relational learning problem settings. We also presented 2 Yale face experiments that requires complex high-level perception reasoning. We hope that our response and added material clearly explains the relevancy and significance of the MNIST examples and why we believe they are difficult to solve with existing methods. In addition, all the experiments are solved with the same principled method which further demonstrated that the proposed VRL method is robust, stable, and generalizable to many different relationships.

We would like to thank the reviewers again for their suggestions on adding more explanation and experiments to the paper; we do believe our latest revision is more readable and self-contained. As we enter stage-2 review, we look forward to continue our discussion and clarify any additional question/concern you may have.

---

### Author Response · Authors · 2020-11-19
**We are here to answer and clarify any questions/concerns!!**

Dear Reviewers and AC:

Stage-2 review is well underway and we are standing by to answer any additional questions/concerns you have.
We have made every effort to answer your initial questions and concerns and hope that you find them satisfactory (please see our response and latest revision including added complex experiments). So far, we do not believe there are outstanding questions about the novelty, validity, and technical correctness of our work. If you have any additional questions or suggestions, please let us know at your earliest convenience so we can address them in a timely manner. We are deeply appreciative of your time and we hope to address all your questions to help you make an informed evaluation.

Thank you,

Authors

---

### Decision · Program_Chairs · 2021-01-07
**Final Decision**

**Decision:**

Reject

**Comment:**

This paper presents a variational learning framework for inferring relations between data points. The authors further introduce novel regularizers to avoid unfavorable solutions to their relational learning problem. Qualitative results are provided on rotated versions of MNIST. Additional qualitative results on the Yale face dataset are provided in the appendix.

The reviewers agree that the idea overall is interesting, and the chosen experiments certainly provide some insight into the idea behind the method and demonstrate that the method is learning reasonable representations of relations between data points. I share the sentiment of the reviewers, however, that this paper is not yet ready for publication. The paper in its current form lacks clear positioning against related problems and related research in this community, and the experiments are all qualitative in nature without the attempt to rigorously compare the proposed method against established techniques. The argument brought forward by the authors that the proposed problem is completely novel and therefore no baselines exist is unconvincing as pointed out by the reviewers, as there is related research on e.g. spatial transformer networks [1], neural relational inference [2], and discovery of causal mechanisms [3], which similarly address the problem of discovering relations, interactions, or transformations. Even if these methods don't exactly fit the problem setup presented in this paper, an attempt should be made to design an evaluation that allows one to compare against some of these approaches, especially given that the paper claims to address the general problem of inferring relations between pairs of data points. Overall, I am confident that this would make the paper stronger and more relevant to this community.

[1] Jaderberg et al., Spatial Transformer Networks (NeurIPS 2015)
[2] Kipf et al., Neural Relational Inference for Interacting Systems (ICML 2018)
[3] Parascandolo et al., Learning Independent Causal Mechanisms (ICML 2018)